# FlyView: a bio-informed optical flow truth dataset for visual navigation using panoramic stereo vision

**Alix Leroy**[*]
Oxford Flight Group,
Department of Biology,
University of Oxford,
Oxford, OX1 3SZ, UK
`alix.leroy@jesus.ox.ac.uk`

**Graham K. Taylor**
Oxford Flight Group,
Department of Biology,
University of Oxford,
Oxford, OX1 3SZ, UK
`graham.taylor@biology.ox.ac.uk`

## Abstract

Flying at speed through complex environments is a difficult task that has been performed successfully by insects since the Carboniferous [1], but remains a challenge for robotic and autonomous systems. Insects navigate the world using optical flow sensed by their compound eyes, which they process using a deep neural network implemented on hardware weighing just a few milligrams. Deploying an insect-inspired network architecture in computer vision could therefore enable more efficient and effective ways of estimating structure and self-motion using optical flow. Training a bio-informed deep network to implement these tasks requires biologically relevant training, test, and validation data. To this end, we introduce *FlyView*[1], a novel bio-informed truth dataset for visual navigation. This simulated dataset is rendered using open source 3D scenes in which the agent's position is known at every frame, and is accompanied by truth data on depth, self-motion, and motion flow. This dataset comprising 42,475 frames has several key features that are missing from existing optical flow datasets, including: (i) panoramic camera images, with a monocular and binocular field of view matched to that of a fly's compound eyes; (ii) dynamically meaningful self-motion, modelled on motion primitives or the 3D trajectories of drones and flies; and (iii) complex natural and indoor environments, including reflective surfaces, fog, and clouds.

## 1 Introduction

Changes in illumination (e.g. shadows, reflections), occlusion (e.g. vegetation, structures), and other sources of large-scale variation (e.g. different substrates, water bodies) present critical challenges to visual navigation in natural and indoor environments. Optical flow is an important visual cue that insects such as flies use to navigate such environments with a brain containing approximately 100,000 neurons and weighing only a few milligrams [2]. Optical flow has also been used by drones for navigation tasks involving estimation of attitude and position, using inefficient and constrained algorithms whose application is typically limited to simple tasks such as holding position in hover. Bio-informed technologies have therefore been suggested for making optical flow algorithms available for small platforms [3–10], potentially reducing the number of onboard sensors required for navigation [11, 12]. These bio-informed approaches are often distinguished by their use of cameras with a very large field of view, which is important both to estimating the structure of the environment, and to inferring self-motion during complex flight maneuvers.

---

[*]Corresponding author
[1]https://github.com/Ahleroy/FlyView

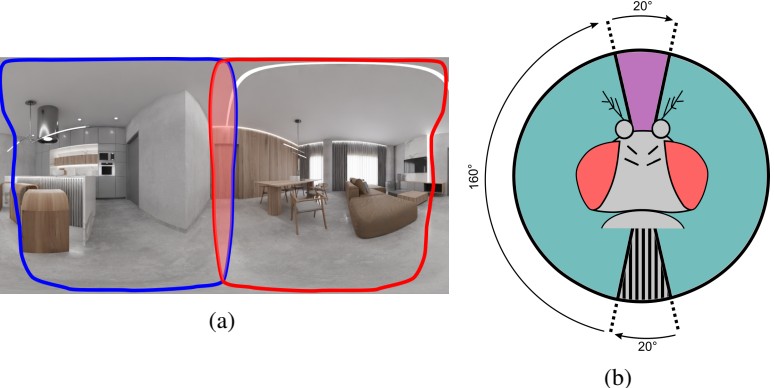

(a)

(b)

Figure 1: (a) Field of view of the blowfly *Calliphora vicina* mapped onto a panoramic view of an indoor scene. Blue and red borders outline the visual field of the left and right eyes; tinted area denotes area of binocular overlap. (b) Schematic representing overhead view of blowfly visual field. Note the 20° sector of binocular overlap (magenta), 20° blind sector (hatched), and the 160° monocular area on each side, giving a total of 340° coverage in the horizontal, which the *FlyView* dataset captures. The dataset includes a binary mask for extracting pixels located within these areas.

Many recent innovations in computer vision have been stimulated by the release of new datasets for machine learning. In particular, large optical flow truth datasets have enabled data-hungry deep learning algorithms to be trained to recognise motion flow, and to be evaluated against well defined benchmarks. Whilst existing datasets capture a range of specific scenarios, including foreground motion seen from the perspective of a stationary camera [13], and motion flow generated by a moving car [14–18], applications to date have focused on estimating dense motion flow maps. Applications to downstream guidance, navigation, and control (GNC) tasks, such as obstacle avoidance, visual odometry, and state estimation, have typically been neglected. Yet, these are all key functions of optical flow detection in insects [19], and it is their performance on these downstream tasks—not the estimation of dense motion flow—that makes insects such remarkable fliers.

These downstream GNC tasks are enabled by the large field of view of an insect's compound eyes (Figure 1). Their almost 360° field of view is not matched by any previously-released optical flow truth dataset for machine learning, but is important to facilitating disambiguation of self-motion stimuli, aided by the embedding of prior information on the insect's natural dynamics in the deep neural network of its motion vision system [19]. Both features are expected to be essential to the successful functioning of deep networks used for visual navigation during complex flight maneuvers in complex environments. This in turn requires the relevant information to be captured in any dataset used for machine learning. To this end, we present *FlyView*, a bio-informed dataset designed to facilitate the training, validation, and testing of new deep networks using optical flow to estimate self-motion in flight.

We generated our *FlyView* dataset synthetically, by rendering open-source 3D scenes in Blender 2.93 (Blender Foundation, Amsterdam) for >100 virtual camera trajectories composed of motion primitives (i.e. pure rotation or translation) or flight trajectories captured for real agents (e.g. quadrotors or insects). This bio-informed dataset makes four main contributions to computer vision:

1. *FlyView* includes panoramic RGB images rendered in both monocular and binocular views. It provides accurate ground truth for several tasks, including forward and backward motion flow fields generated by the camera's self-motion through the scene, accompanied by the depth map for all camera views.

2. *FlyView* is designed with downstream GNC tasks of flying vehicles in mind. The camera therefore flies through a photo-realistic indoor or outdoor environment, and its pose is known for every frame, permitting use of the dataset for tasks involving self-motion estimation.

3. *FlyView* is, to our knowledge, the first optical flow truth dataset to include motion dynamics relevant to flight. It contains a range of dynamically meaningful motions, including the flight trajectories of small quadrotors and blowflies, and simple motion primitives involving different axes of rotation and translation.

4. *FlyView* is the second largest optical flow dataset in terms of the number of instances it contains (42,475 frames from each of three virtual cameras, in 9 new and diverse indoor and outdoor environments). It is available open-source, and is designed to be compatible with existing computer vision approaches, offering new and complementary data for the training and evaluation of computer vision algorithms.

Whilst *FlyView*'s field of view, stereo baseline, binocular overlap, and sampling resolution are matched to those of a fly (Figure 1), the data are presented in a conventional rectangular pixel array from the perspective of a high-acuity omnidirectional camera. The images may be post-processed to model the optical input to the low-acuity hexagonal imaging array of the compound eye of an insect if desired, but this is not done here in the interests of generality. For applications in self-motion estimation, the synthetic data described in the main paper are supplemented by a smaller set of real video data from a pair of divergent stereo fisheye cameras undergoing pure linear translational motion indoors. As described in the Supplementary Materials (Section H), these real video data are accompanied by accurate truth data on the camera's self-motion, but lack the motion flow ground truth that is a key feature of *FlyView* itself. We demonstrate the value of *FlyView* for computer vision by using it to evaluate the performance of a state-of-the-art pre-trained RAFT network [20]. This analysis shows the practical difficulties associated with applying networks trained on narrow-field of view images to omnidirectional camera data, motivating the future use of *FlyView* in training of optical flow algorithms.

## 2 Related works

A large number of natural and synthetic optical flow truth datasets have been created in recent years. These have enabled computer vision researchers to implement motion flow estimation, to highlight limitations of current methods, to develop more robust solutions, and to challenge existing benchmarks. Here, we review the datasets that have been used in optical flow algorithm evaluation and model training to date, highlighting their limitations for self-motion estimation. Identifying accurate ground truth motion flow in natural scenes is a challenging task [14–16], but extracting synthetic ground truth from simulations has proven a flexible and efficient approach [13, 17, 18, 21–23], allowing fine variation in scene parameters, and resulting in accurate and dense ground truth compared to analyses of real video data. Recent datasets contain specific motions and environments that simulate important challenges, including large displacements, scene occlusion, and varied illumination, all matching real-world scenarios.

The full range of available optical flow datasets is summarised in Table 1 and reviewed briefly here. The *Middlebury* dataset [21] offered an interesting first dataset for benchmarking, but *MPI-Sintel* [22] was the first to be generated using synthetic scenes extracted from pre-existing 3D assets. Concerns regarding the non-natural aspect of the images of *MPI-Sintel* and other datasets that followed [13, 23] led to datasets specializing in naturalistic visual images. For instance, *VKitti* [17] and *VKitti2* [18] contain synthetic images based on the natural scenes of the original *Kitti* dataset [14, 15], which are enhanced with a real-to-virtual-world cloning method to look natural while providing accurate and complete ground truth. These standard datasets have proven fertile ground for the training and evaluation of computer vision models for motion flow estimation. Indeed, thanks to the collective size of these datasets, modern data-based algorithms now achieve state-of-the-art results outperforming classical methods on every metric [20]—even computational time for mobile-oriented deep networks.

Regardless of their utility in training algorithms to perform dense motion flow estimation tasks, standard datasets suffer from several key limitations when applied to self-motion estimation. Most of these datasets comprise views from static or slow-moving cameras with dynamic scenes, often involving unrealistic motions such as those contained in the *FlyingChairs* dataset (Table 1). This lack of dynamically-meaningful motions means that these datasets contain motion flow fields which poorly represent the dynamics of maneuverable vehicles in real world scenarios. Datasets collected in autonomous driving scenarios are partial exceptions to this rule, with optical flow fields reflecting the vehicle's self-motion [14–18]. However, these mainly contain forward motion of a forward-facing camera, thereby limiting the resulting optical flow fields to one simple self-motion scenario. Other accurate datasets with a large span of annotations, such as SAMA-VTOL [24], in principle allow the user to reconstruct a motion flow map from dense point-clouds and semantic maps, but do not contain motion flow truth data at present.

Table 1: Summary of key features of existing optical flow truth datasets.

| Dataset | Scene | Dense Groundtruth | Backward Flow | Depth | Stereo | Extrinsics | Total Scenes | Training Frames | Test Frames | Total Frames | Resolution (pixels) |
|---|---|---|---|---|---|---|---|---|---|---|---|
| UCL [31] | synthetic | ✓ | | | | | 4 | N/A | N/A | 4 | 640×480 |
| UCL (extended) [31] | synthetic | ✓ | | | | | 20 | N/A | N/A | 20 | 640×480 |
| Middlebury [21] | both | ✓ | | | | | 12 | 84 | 90 | 174 | 640×480 |
| KITTI2012 [14] | natural | | | ✓ | ✓ | ✓ | 194 | 194 | 195 | 389 | 1226×370 |
| KITTI2015 [15] | natural | | | ✓ | ✓ | ✓ | 200 | 200 | 200 | 200 | 1242×375 |
| KITTI Virtual [17] | synthetic | ✓ | ✓ | ✓ | | ✓ | 5 | 21,260 | 0 | 21,260 | 1242×375 |
| VKITTI2 [18] | synthetic | ✓ | ✓ | ✓ | ✓ | ✓ | 5 | 21,260 | 0 | 21,260 | 1242×375 |
| MPI-Sintel [22] | synthetic | ✓ | | ✓ | | ✓ | 25 | 1,064 | 564 | 1,628 | 1024×436 |
| FlyingChairs [13] | synthetic | ✓ | | | | | 964 | 22,232 | 640 | 22,872 | 512×384 |
| FlyingThings3D [23] | synthetic | ✓ | ✓ | ✓ | ✓ | ✓ | 2,247 | 21,818 | 4248 | 26,066 | 960×540 |
| Monka [23] | synthetic | ✓ | ✓ | ✓ | ✓ | ✓ | 8 | 8,591 | 0 | 8,591 | 960×540 |
| Driving [23] | synthetic | ✓ | ✓ | ✓ | ✓ | ✓ | 1 | 4,392 | 0 | 4,392 | 960×540 |
| SceneNet RGB-D [32] | synthetic | ✓ | | ✓ | | ✓ | ~17,000 | ~5,000,000 | ~300,000 | ~5,000,000 | 320×240 |
| HD1K [16] | natural | ✓ | | ✓ | | ✓ | 36 | 1,083 | 54 | 1,137 | 2560×1080 |
| SynWoodScape [30] | synthetic | ✓ | | ✓ | ✓ | ✓ | not available | N/A | N/A | ~80.000 | 1280×966 |
| FlyView (this paper) | synthetic | ✓ | ✓ | ✓ | ✓ | ✓ | 9 | 29,975 | 12,550 | 42,475 | 1700×900 |

Most standard optical flow truth datasets use a single camera with a narrow field of view. This is limiting in self-motion tasks, for which a large field of view is necessary to eliminate ambiguity in the local optical flow field [25]. Indeed, in some specific scenarios involving narrow field of view cameras, the direction of motion may lie outside the camera field of view, leading to self-motion estimation that is extremely sensitive to noise [26]. Despite some important advantages in using wide angle cameras for navigation tasks [27], and with few drawbacks for lenses up to 180° field of view, large field of view cameras have remained almost absent from optical flow truth datasets to date. Recently, a new driving dataset named *Kitti360* was collected from two equirectangular cameras covering 360° field of view [28], but a planned update including motion flow ground truth is still missing at the time of writing. Another four-camera fisheye surround-view dataset for autonomous driving called *WoodScape* [29] was recently extended for motion flow estimation with the creation a synthetic omnidirectional dataset called *SynWoodScape* containing motion flow ground truth among other annotations [30]. *SynWoodScape* had not been fully released at the time of writing.

Apart from the limitations inherent to existing computer vision datasets, no analogous dataset could be found containing input luminance, motion flow, motion state, and depth data for a flying insect. Such data could enable important breakthroughs in modelling the visual system of insects, including modelling the response of specific neurons, and offering better understanding of their mechanisms of optical flow computation. For these reasons, we decided to design and collect our own synthetic dataset matching bio-informed self-motion estimation requirements.

# 3   Methods

State-of-the-art algorithms for motion flow estimation rely on data-hungry deep networks. Not only is a large volume of data necessary to train a model; additional data is also required to validate the overall training strategy, and to test the model's performance. These three different stages of training, validation, and testing usually make use of three non-overlapping subsets of data sharing a similar underlying distribution of information. Together, these three subsets must encompass a large quantity of data that represents as accurately as possible the real-world scenario that the agent will face. The more realistic the dataset, the better the network will perform in real-world scenarios. In light of our analysis of existing standard datasets and their limitations (Section 2), we decided to design and generate a dataset for bio-informed visual navigation that would also be compatible with existing computer vision approaches.

## 3.1   Dataset requirements

To maximise compatibility, we require our optical flow truth dataset to comprise rectangular images composed of square pixels, rather than outputs modelling the hexagonal sensor array of an insect's compound eyes. We define five biologically-motivated requirements for our data, analogous to the autonomous driving requirements defined previously [16]:

R1   Images with a large field of view matching an insect's compound eyes. Besides affording biological realism, this is important in avoiding ambiguities in self-motion estimation [25], guaranteeing that the axes of rotation and translation fall within the field of view.

R2 Stereo images with binocular overlap modelled on an insect's compound eyes, allowing downstream fusion of input from the two visual hemispheres. Binocular vision offers continuity across the midline of the visual field, reduces uncertainty in complex scenarios, and in principle enables stereo depth estimation within the area of binocular overlap.

R3 Dynamically-meaningful self-motions modelled on the flight trajectories of flies and other agents such as quadrotors, plus simpler motion primitives corresponding to pure translation or rotation, without the complexities of coupled rotational and translational motion flow.

R4 Photo-realistic indoor and outdoor environments allowing functional discrimination of dorsal motion flow from ventral motion flow. The latter is generally more useful for self-motion estimation, as reflected in the matched filters of the fly motion vision system.

R5 A large array of photoreceptors offering a sampling resolution sufficient to model the compound eyes of insects, which may contain from a few hundred to tens of thousands of discrete photoreceptive units called ommatidia.

We respond to these requirements by describing our ground truth generation method for a synthetic camera system modelled specifically on the visual anatomy of flies (Section 3.2). We then describe the flight trajectories used to sample the virtual environments generating the data (Section 3.3).

## 3.2 Camera rendering method

We used open-source 3D photo-realistic synthetic scenes, which we rendered using the Cycles engine in Blender (Figure 3). This 3D render engine generates visual images of the scenes using ray tracing. Predefined flight trajectories (see below) were loaded onto the Blender key frames using a customer-made Python script interacting with the Blender API. The scenes were selected for the range of environments they covered, including indoor and outdoor environments (R4). These include both bright and dark scenes covering a range of volumes, with varying illumination, occlusion, and miscellaneous optical effects including fog and specular reflection (see Supplementary Materials, Section A). The camera setup that we used to render the scenes comprised three virtual cameras, each modelled using the equirectangular model officially supported by Blender. This approach allowed us to cover the complete field of view of a fly (R1), and to represent the motion flow field over a similar range of azimuth and elevation to that represented in a fly's measured response to widefield optical flow stimuli [33].

We set up one pair of cameras to represent the compound eyes of either *Drosophila melanogaster* [34] or *Calliphora vicina*, with a horizontal and vertical field of view of 180° in each eye. The principal axes of the cameras were oriented 160° apart (see Figure 1), giving a biologically-realistic field of view of 340° in the horizontal, with 20° of binocular overlap (R2). The cameras are separated by a 4 mm stereo baseline, modelled on measurements performed on adult female *Calliphora* which are several times larger than *Drosophila*. Typical stereo vision systems have a baseline from tens of centimeters (HD1K 0.3m, KITTI 0.54m) to more than one meter (FlyingThings3D, Monkaa, Driving), enabling accurate depth extraction over distances of tens of metres. In contrast, the 4 mm baseline between the compound eyes of the fly is too small for the binocular disparity to be used to extract useful depth information, except at distances much less than 1 m. This baseline displacement may nevertheless be relevant when modelling downstream fusion of binocular information in flies (e.g. when analysing structure-from-motion), so is retained accordingly.

The third camera was used to render images with a monocular field of view equivalent to that of the camera pair, allowing almost complete visualization of the scene in one panoramic view (R5). This approach allows us to mimic the fly's 340° field of view, without the issues of discontinuous motion flow that arise with a 360° virtual camera in Blender. This single panoramic camera captures a $1700 \times 900$ pixel image, whereas the two cameras in the pair each capture a $900 \times 900$ pixel image. Each camera thereby samples 5 pixels per degree in the horizontal and vertical directions at the centre of the image, with an even higher sampling resolution towards the edges of the images. This sampling resolution exceeds the typical angular resolution of *Drosophila* and *Calliphora*, which are estimated to sample 0.2 or 2 photoreceptors per degree, respectively. The high sampling resolution of the images in our *FlyView* dataset therefore allows it to be used to model a wide range of insects having a similar field of view, and offers fine-grained ground truth for computer vision applications.

We used Blender to extract the RGB renderings, motion flow, and depth map for each camera, at a frame rate appropriate to the motion dynamics (see Section 3.3 below). The images and other Blender

passes were generated using two NVIDIA RTX 3090 GPUs. This computational power allowed us to generate high resolution truth data for tens of thousands images over a period of a few months. Motion flow was retrieved directly from the projection of the pixel's 3D position onto the image sensor, and the depth map was generated by computing the distance between the pixel's position and the scene along the corresponding ray. Additional information related to camera positioning, sampling resolution and field of view can be found in the Supplementary Materials (Section E).

### 3.3    Trajectory generation

To diversify the dataset (R3) and enable better understanding of visual navigation tasks, we modelled three distinct classes of motion. The first class of motion comprises simple motion primitives (Section 3.3.1), being composed of either pure translational or pure rotational motion. This is important to modelling the different kinds of optical flow field that are generated by these two distinct kinds of motion, without the additional complexities that arise from their coupling. The second class of motion was generated by tracking a small quadrotor in flight (Section 3.3.2). The resulting motions are composed of comparatively simple combinations of rotation and translation, made more complex by the drone's flight dynamics. The third class of motion was generated based on previous work [35] tracking *Calliphora* flies using high-speed video photogrammetry (Section 3.3.3). Each class of motion is described below, and summarised in the Supplementary Materials (Section B.2).

### 3.3.1    Motion primitives

Pure rotational motion flow is invariant to the structure of the environment, so rotations around the same camera-body axis always result in the same characteristic motion flow field. In contrast, translational motion flow depends on the distance from the observer to every point in the observed scene. In order to highlight the differences between rotational and translational self-motion, and to account for the possibility that gaze may be stabilized rotationally during the flight maneuvers represented by the other two classes of motion, we included pure translational and rotational trajectories in the dataset. We generated these trajectories as constant velocity or constant angular velocity motions, in directions sampled along the principal axes of the visual system, and uniformly along intermediate axes on the unit sphere. The trajectories were generated with the assumption of camera height not exceeding 2 m above the ground, and usually began with the camera system level, such that the resulting dataset contains meaningful prior information on the distribution of luminance and structure within a typical scene. Some specific translational motions were repeated in multiple scenes, as the same translational motion leads to different motion flow in different scenes. The angular speeds of the motions ranged up to $36°$ s$^{-1}$, and we used a 25 Hz sampling rate, chosen to match the frame rate of a basic video camera.

### 3.3.2    Drone trajectories

We designed the quadrotor trajectories (Figure 9) to complement these motion primitives, by including time-varying velocities and/or angular velocities in translational and/or rotational motion. Using a DJI Tello drone, we were able to produce simple motions such as pure translation or rotation, whilst also aggregating some complexity on the dynamics resulting from the vehicle's stability and control. For example, in order to perform forward translational motion from an initial condition of stationary hover, the drone will pitch nose-down and throttle up so that its rotor discs can push it in the desired direction. A similar pitching mechanism is used by flies to control the elevation of their flight path [36], so the kinds of motion coupling that the drone dataset contains are typical of those experienced in rotary-wing and flapping-wing flight. Other couplings would be present in fixed-wing vehicles, and could in principle be exploited in training a deep network to estimate their self-motion.

The drone was equipped with retroreflective markers fixed onto its fuselage and rotor guards (Figure 2) and was tracked using a motion capture system (MoCap) with 22 Vicon Vantage V16 cameras sampling at 200 Hz [37]. This sampling frequency is comparable to the flicker fusion frequency of a blowfly *Calliphora* [38], so is expected to represent all relevant frequencies of the drone's motion dynamics. We used the same 200 Hz sampling rate for the renderings, thereby capturing all of the motion information contained in the MoCap trajectories. The flight trajectories were realized by using the official DJI Tello application to control the drone. This offers a range of pre-defined motions and actions such as take-off, landing, circling around a specific point, or even directional flips. Most of these pre-defined motions were straightforward for the MoCap to track, but some involved complex

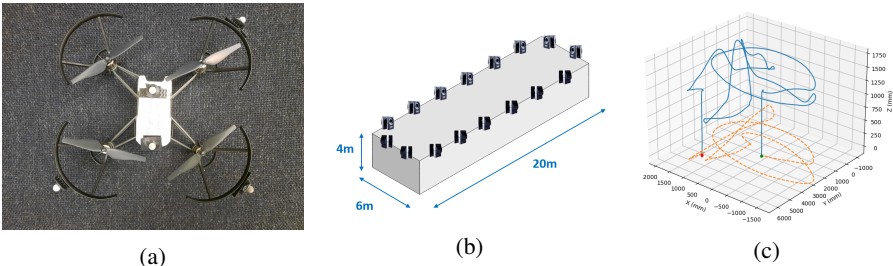

|  |  |  |
|:---:|:---:|:---:|
| (a) | (b) | (c) |

Figure 2: (a) Top view of a DJI Tello drone with retro-reflective markers to enable tracking. Two markers were fixed on the fuselage, and another three markers were fixed on the protective structure to maximize the distance between them. (b) Schematic of the flight arena in which the drone trajectories were collected with a sub-millimeter accuracy covering the entire flight volume.(c) Example of a free-flight drone trajectory (blue) captured with the MoCap and replicated in the *FlyView* dataset. The trajectory includes both take-off (green point) and landing motions (red point). The trajectory is projected to the ground (orange) for visualization.

maneuvers that were more challenging to track. For instance, the flip motion leads to fast and unstable flight maneuvers, reducing the visibility of the markers because of possible motion blur and occlusion of the markers when the drone is inverted. We used the manual mode to generate other motions such as translation or yaw rotation, so the resulting trajectories display some further user-generated complexity beyond pure translation or rotation.

### 3.3.3 Fly trajectories

To capture the even more complex motion dynamics of a fly, we used flight trajectories that had been reconstructed previously for *Calliphora* filmed flying inside a 1 m diameter spherical arena [35]. These trajectories involve flight at speeds up to 1.5 m s$^{-1}$, and describe the coupled rotation and translation of the fly's body, as measured using video photogrammetry [35]. They neglect the rotation of the fly's head with respect to its body, and therefore ignore the effect of gaze stabilization. Flies stabilize their gaze by rotating their head around their body's longitudinal axis in order to compensate body roll during flight [39]. Including head compensation would therefore remove rotational optical flow components that, even if minimized by the fly, are detected in the fly's final layers of its visual system [19]. Whilst removal of the rotational components of the optical flow field may be important in simplifying detection of its translational components, this couples the dynamics of flight with the dynamics of gaze stabilization. We therefore decided to use the body's pose to define the camera's pose, but leave open the possibility of adding head stabilization at a later date. We sub-selected 36 of the 257 flight trajectories that had been collected previously for *Calliphora* [35], covering a diverse range of flight maneuvers (Figure 9). The trajectories were converted from the Earth-fixed reference frame in which they were reported (with orientation expressed by a set of intrinsic Euler angles) to Blender's reference frame, using a quaternion to describe body orientation. The fly trajectory data were sampled at a rate of 100 datapoints per wingbeat, which is nearly two orders of magnitude higher than the flicker fusion frequency of the fly [38]. Although the dataset therefore contains temporal frequencies higher than the fly can sense, we chose to render the images at the same sampling frequency to avoid any loss of data, and to allow subsequent temporal filtering of the luminance signal to model the dynamical response of the insect's photoreceptors if so desired.

## 4 Results

In this section, we provide a brief overview of our *FlyView* dataset (Section 4.1), before comparing this to the existing datasets (Section 4.2) described in Table 1. Further quantitative data analysis can be found in Section 4.3 and in the Supplementary Materials.

### 4.1 Overview of dataset

*FlyView* is a large and diverse dataset containing 42,475 frames rendered for each of three cameras defined to provide a 340° panoramic view of the environment (see Section 3.2). These frames

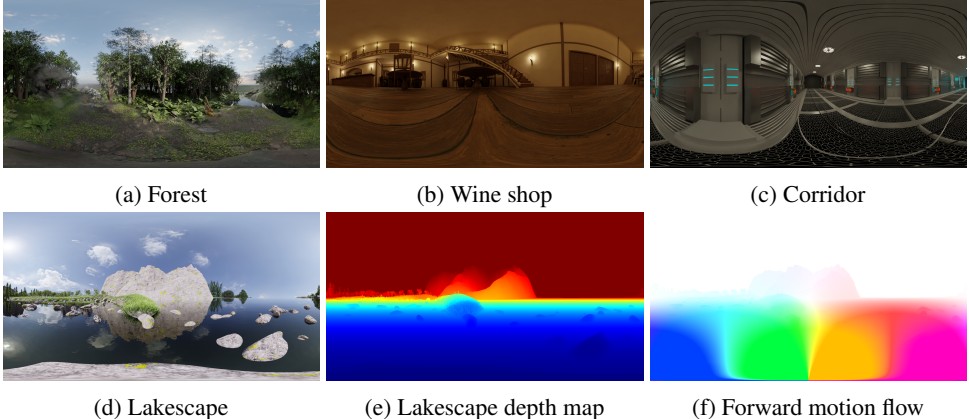

(a) Forest            (b) Wine shop           (c) Corridor

(d) Lakescape      (e) Lakescape depth map      (f) Forward motion flow

Figure 3: Examples of data contained within *FlyView*. (a-d) Examples of scenes used for data generation; for details of authorship see the Supplementary Materials (Section D). (e) Depth map associated with the visual image in (d). (f) Motion flow during forward translation through (d).

record only static scenes, avoiding any flight direction ambiguity [25] and permitting analysis of visual navigation in the absence of noise. We provide RGB images in a lossless 16-bits PNG format permitting the common 8-bits compression used by numerous computer vision algorithms, and the analysis of the RAW luminance signal captured by the sensor. The corresponding depth map and motion flow field is provided in a 32-bits format, offering a fine description of the environment. We provide the intrinsic and extrinsic parameters of the camera system for all frames and views, including the Cartesian position and quaternion orientation of each camera as required for assessing self-motion and structure-from-motion tasks. In addition, we provide a set of calibration images containing checkerboards, allowing the user to recalibrate the intrinsic parameters using a different camera model to the equirectangular model that we used.

When training a deep network, one of the main challenges is in dealing with over-fitting, which occurs when the model specializes on a specific dataset, but cannot accurately infer the results on data with a different distribution. In other words, the model fails to generalize its task. Whilst over-fitting can be avoided with specific deep learning methods, it can also be addressed at an earlier stage by designing the dataset appropriately, as we have done here. The different types of motion that we model result in different image views and different motion flow fields that will help avoid over-fitting. Another way of avoiding overfitting is to vary the environment. Most existing datasets (Section 4.2) therefore make use of numerous environments to increase their size and diversity. In the same way, our dataset is composed of 9 different scenes that provide a wide distribution of visual images (Figures 5). Our scenes span from small to large 3D environments, both indoor and outdoor. They contain different illumination effects under natural and artificial lighting, including light reflection on water and the effects of fog. The 3D scenes also contain dark and light areas, creating challenging environments for visual navigation. Finally, we suggest a suitable split of the data into training, validation, and test sets aimed at maximizing the exploitation and distribution of features in the scenes such as outdoor/indoor locations, bright/dark scenes, and small/large displacements. This dataset split is detailed in the Supplementary Materials (Section C.4).

## 4.2 Dataset comparison

Table 1 compares our *FlyView* dataset to existing optical flow truth datasets across a range of metrics. *FlyView* is distinguished by the high resolution of its images, by its wide field of view, and by its large number of frames. Because of its biological inspiration, *FlyView* falls into its a category of its own, but was designed to be compatible with existing computer vision approaches. In particular, the data were generated using virtual cameras that output dense rectangular matrices of visual images and ground truth, in a similar fashion to other datasets feeding computer vision algorithms. *FlyView* is therefore a prime candidate for evaluation and benchmarking of the performance of different algorithms, as well as providing an excellent complementary set for training machine learning models. Most notably, its bio-informed design is specifically adapted to drive innovation around visual navigation systems based on large field of view cameras.

Table 2: Comparison of performance in motion flow estimation (mean $\pm$ s.d. of angular and magnitude errors) of the pre-trained RAFT-large network on test data from *FlyingChairs* and *FlyingChairs3D* versus various subsets of *FlyView* (see Supplementary Materials, Section C.3).

| | FlyingChairs | FlyingThings3D | FlyView S1 | FlyView S2' | FlyView S3 | FlyView S5 | FlyView S8 |
|---|---|---|---|---|---|---|---|
| **angular error** (°) | 6.42 $\pm$7.09 | 4.11 $\pm$3.77 | 6.90 $\pm$6.10 | 18.83 $\pm$4.79 | 13.13 $\pm$8.36 | 18.62 $\pm$8.87 | 18.43 $\pm$9.61 |
| **magnitude error** (px) | 1.26 $\pm$1.37 | 4.23 $\pm$6.28 | 6.72 $\pm$5.69 | 0.69 $\pm$0.68 | 4.67 $\pm$5.62 | 3.06 $\pm$3.66 | 0.93 $\pm$0.77 |

One of the most important distinguishing features of *FlyView* is its large field of view, which results in almost complete motion flow maps describing both small and large apparent motions. The wide range of motion flow that can be present within a scene is highlighted in Figure 3f for one of the outdoor scenes. In this case, self-motion can be estimated most reliably using ventral optical flow, given that the sky produces little dorsal optical flow. This is a prior that is known to be built in to the neuronal weightings of the fly's visual system [33], and is therefore worthwhile to capture in a dataset aimed at training deep networks for self-motion estimation. Additionally, many standard datasets lack well-defined validation and test sets. Some provide visual images without motion flow ground truth (e.g. MPI Sintel), others do not provide any defined test set (e.g. Monkaa and Driving), or any defined validation set (e.g. FlyingThings3D, FlyingChairs). In contrast, FlyView has been designed to include both a validation and a test set with accurate ground truth. Appendix C provides a detailed description of our recommended split.

### 4.3 Quantitative and qualitative analysis

We evaluated the performance of a state-of-the-art RAFT network [20] on a diverse range of subsets of *FlyView*, having pre-trained the network on *FlyingChairs*, *FlyingThings3D*, *HD1K*, *MPI-SIntel*, and *KittiFlow*. We compared its performance on *FlyView* to test sets from *FlyingChairs* and *FlyingThings3D*, for both the large and small versions of the RAFT network defined in [20]. Table 2 shows that even the RAFT-large network fails to accurately and homogeneously compute the motion flow map on *FlyView*, with an unacceptably high mean angular error in some scenarios. The small magnitude error in *FlyView* sets $S2'$ and $S8$, defined as the error on the norm of the vector field, can be explained by the small displacement of the camera in these sets. Qualitatively speaking, RAFT has difficulties in estimating the magnitude of the motion flow vector close to the edges of the images, where the distortion is important. This is because the pixel angular density of the image is much higher in these regions, leading to more significant apparent motions for a slight displacement of the camera. As described in Section 4.2, ventral optical flow is of utmost importance for self-motion estimation in flight over the ground, so RAFT's failure to estimate the motion flow accurately in this part of a wide field of view image presents a significant barrier to accurate self-motion estimation.

Similarly, RAFT has difficulties in correctly estimating the motion flow of objects close to the camera, for which the apparent motion is fast, and for which the effects of image distortion are large. Additionally, RAFT fails to estimate the motion flow vector's angle accurately when the apparent motion is small. As the angular density of the pixels is smallest in the central part of the image, apparent motion can be very small in this region, causing RAFT to fail dramatically in this specific scenario. The smaller version of the network, RAFT-small, is very sensitive to the noise introduced by ray-tracing during rendering (see Figure **??**). This makes estimation of the angle of the motion flow vector noisy, particularly in respect of large flat surfaces such as walls and ceilings. As a result, because the RAFT network was pre-trained on data largely without distortions, and with a reasonably homogeneous pixel angular density in the entire image, it fails in the more general scenarios encountered throughout the *FlyView* dataset, highlighting the need for optics-specific datasets for training. Optionally, modifications to existing network architectures can be achieved to allow a more accurate estimate of optical flow on omnidirectional images [40, 41].

## 5 Conclusion

In this paper, we designed and collected the first bio-informed dataset for motion flow and self-motion estimation. This dataset is one of the largest motion flow datasets available publicly online, and one of only a few to offer large sensor resolution. To the best of our knowledge, *FlyView* is the largest motion flow dataset with meaningful and varied dynamics, and the first dataset to focus on translational and

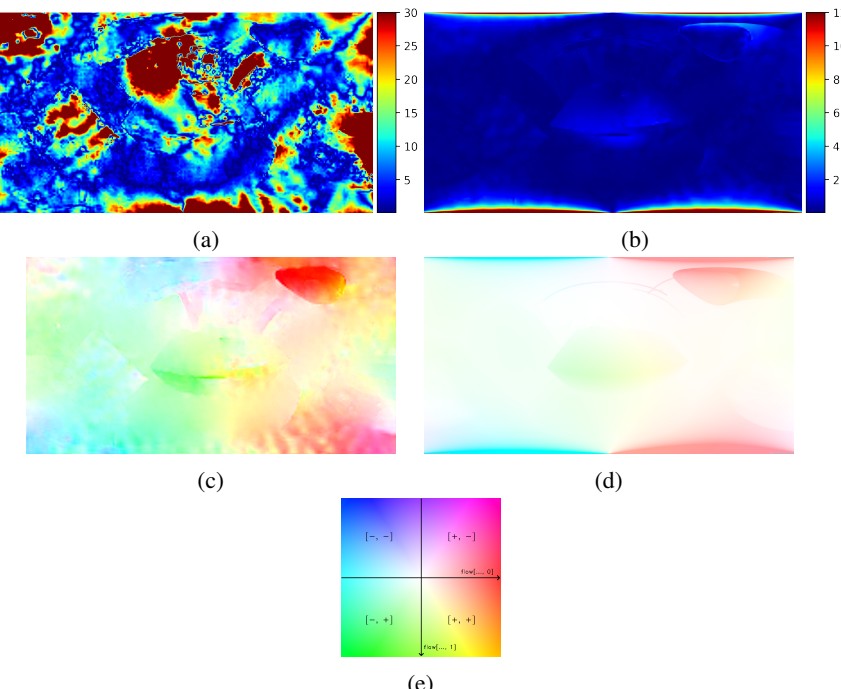

Figure 4: Error in motion flow estimation by the pre-trained RAFT networks tested on *FlyView*. (a) Heatmap of the angular error ($°$) of the RAFT-large network, where red denotes errors $> 25°$; (b) Heatmap of the magnitude error (px) of the RAFT-large network, where red denotes errors $> 10$ px; (c) Noisy motion flow estimation by the RAFT-small network; (d) Motion flow truth data for (c). For further examples, see Supplementary Materials. (e) Definition of the flow

rotational motions independently and in combination. The *FlyView* dataset relies on a camera setup based on measurements on flies, and therefore offers an almost complete panoramic field of view for its visual images, motion flow, and depth maps. This camera setup, mimicking the visual field of an insect, brings new challenges and opportunities to biology and computer vision. It is designed to be compatible with existing computer vision approaches, whilst also permitting future biological studies simulating the input to the fly's hexagonal imaging array using the image processing mask provided. The *FlyView* dataset is fully accurate thanks to its synthetic generation of motion flow ground truth, and offers a wide range of photo-realistic visual images in indoor and outdoor environments that require different feature extraction strategies for self-motion estimation. Future work will use these data to train novel bio-informed computer vision algorithms, and to provide a new benchmark for existing state-of-the-art algorithms. A detailed documentation can be found in the GitHub repository of this project.

## Acknowledgments and Disclosure of Funding

AL is supported by a doctoral studentship funded by Dstl, the Department of Biology, and Jesus College, Oxford. This project has received funding from the European Research Council (ERC) under the European Union's Horizon 2020 research and innovation programme (grant agreement No. 682501). We thank Alexander Borst and Holger G Krapp for their useful recommendations and insighful discussions on fly vision. We thank Stuart Golodetz for his recommendations on the split of the dataset, and Inés Dawson and Simon Walker for access to the fly trajectory data. We thank the authors of the 3D scenes for releasing these open-source (see Appendix for the licences and access details for these assets). We declare no competing interests. This document is an overview of UK MOD's Defence Science and Technology Laboratory (Dstl) sponsored research and is released for informational purposes only. The contents of this document should not be interpreted as representing the views of the UK MOD, nor should it be assumed that they reflect any current or future UK MOD policy.

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
