# OpenReview forum: "FlyView: a bio-informed optical flow truth dataset for visual navigation using panoramic stereo vision"
_NeurIPS.cc/2022/Track/Datasets_and_Benchmarks — NeurIPS 2022 Datasets and Benchmarks _

### Official Review · Reviewer_6218 · 2022-07-01
**Review for FlyView**

**Rating:** 6
**Confidence:** 4
**Clarity:** Yes

**Strengths:**

Overall the paper is well motivated (e.g. the problematic has direct applications and the use of large field of view cameras is justified by the authors when they explain that it ensure the motion vector is present in the field of view of the camera), and well written (i.e., pleasant to read). The dataset has no equivalent in prior art and would be useful in practice.

**Weaknesses:**

The major weakness of this work is that all scenes are static (see my comment under Correctness for more details).

Two other points that bother me, is that no evaluation of any kind is conducted on the dataset and no real data is acquired.

The authors claims on line 34 that the relevant metrics should be on downstream tasks. While getting the ground truth optical flow for real images is unpractical, generating the ground truth for a downstream task, e.g. egomotion, should be easier. This should enable the generation of real testing data.

If such real test data was generated, it would be possible to show the benefits of the proposed dataset for training, which is not done at the moment.

Overall, since this paper is submitted to a data specific track, these weaknesses are not game breaker, as the proposed dataset still have clear benefits over the state of the art, but they definitely makes that paper borderline.



**Additional Feedback:**

I find the comparison to the brain of a fly, used on lines 3 to 5 and again on line 21 to be completely uninformative for computer scientists. For comparison with artificial neural networks (which, as far as I know, being software do not weight anything), it might be better to report the number of neurons in a fly's brain (which is apparently 100'000, if I believe the claim in "A Complete Electron Microscopy Volume of the Brain of Adult Drosophila melanogaster [ https://www.cell.com/cell/fulltext/S0092-8674(18)30787-6 ]"), as the number of neurons can be compared with the number of "neurons" in an artificial neural network.

A reference might be needed to support the sentence between lines 21 and 23 about the use of optical flow for drone egomotion but this is a minor issue.

One line 61 the authors claim that their dataset is exceptionally large: this is wrong. While the dataset rank second in therm of number of images, it has a limited number of scenes (only 9, even if the kind of scenes seems more diverse than in other works). I would reformulate that statement to something a bit more objective, underlining the fact that the dataset has a large number of images, spread across a limited number of diverse scenes. It is also important to underline that the scenes are realistic, which is not always a given for simulated datasets.

Finally I was wondering if the authors used the cycles or eevee rendering engines, and if both would be fine or if certain limitations impose the use of cycles ? Eevee is generally much, much faster which is highly beneficial when generating a large amount of simulated images.

Overall, I feel that this work is clearly borderline paper. Some limitations are pretty severe but the dataset is well motivated, has no strict equivalent in pre-existing datasets and would be useful in practice. I would surely upgrade my review to something more positive if the authors were to add at least one scene with moving rigid objects and one scene with moving non-rigid objects, those would be really useful for evaluation and are a cases where the use of simulated data is far more justifiable.

**Correctness:**

The authors used simulation to create their dataset, claiming that real acquisition methods are complex and unprecise. I mostly agree with that statement, but there is one details of utter importance for which I disagree: this is that using a technique like the one used for the Kitti dataset for static scene when subpixel accurate optical flow is not required is still sound and valid.

Now the point is, the proposed dataset contains only static scenes, and the targeted application (i.e. egomotion) do not require subpixel accurate optical flow. In my opinion this is the major weakness of this work. It would be highly desirable that the authors add at least one scene with rigid moving object and one scene with non-rigid moving objects, at least for evaluation purposes.

**Documentation:**

# documentation

The github code is documented.

I found no documentation about the dataset itself. As far as I am concerned, none is required, but I am pretty familiar with blender myself. It might be a good idea to add a ReadMe, maybe even within the blender text editor of the data files, explaining the different objects, naming conventions, and especially which part of the blender file should be left untouched for the processing scripts further downs in the pipeline and which parts can be modified by the final users if they want to create more data in the future.

# maintenance plan

While the authors did not upload the whole dataset at the time of review, they did present a convincing long term storage plan that should ensure the data is still available in the future (i.e., with the data in Zenodo, which is managed by CERN and thus unlikely to go away anytime soon, and the code on github, which has proven a reputable plateform for code starage).

# data format

The sample provided shows dataset will contain both the source .blend files and the images in .exr (to store the ground truth depth and flow passes) and .png format (for the actual views). All of these formats are open, documented formats that are easy to work with in python and c++ and can be opened by freely available software, limiting the barriers for the usage of the data. I noticed that while the authors used the multiview system integrated in blender, they still provided 3 different .blend files which might be unoptimal.

The code make reference to .csv trajectory files but none has been provided with the example, even if the github repository states that a trajectory should be provided. I assume this is a mistake and that the trajectories will be provided with the final dataset.

**Relation To Prior Work:**

The relevant literature has been covered and the differences between previous works and this work have been highlighted.

**Summary And Contributions:**

The authors propose a bio inspired simulated dataset for optical flow based egomotion estimation.

The dataset has been generated from 9 static scenes and contains 42'475 images triplets (each triplet containing 3 cameras). Different motion models have been used including basic motions like pure drifts or pure rotations, as well a natural motions of an UAV and a fly trajectory.

---

> ### Author Response · Authors · 2022-08-22
> **Response part 1**
>
> We thank the reviewer for their time taken in reviewing our work, and for their valuable comments which have helped improve the paper. We have addressed all of their comments as explained in our response below.
>
> ## Weaknesses
>
> The reviewer is correct to note the limitation that all of the scenes in FlyView are static, which we discuss further below. The reviewer is also correct to state that: (1) no evaluation is conducted on the dataset; and (2) no real data is acquired. Concerning these important points:
>
> (1)	We agree that evaluation of the dataset in an important requirement. To tackle this concern, we have added a new evaluation section testing the performance of the state-of-the-art RAFT network when pre-trained on standard computer vision datasets (FlyingChairs, FlyingThings3D, HD1K, Kitti flow, Sintel) and tested on FlyView. This new section demonstrates that the pre-trained RAFT network performs poorly when evaluated on FlyView, showing the importance of taking account of image distortion when applying state-of-the-art optical flow learners to widefield images. In addition, we offer a qualitative analysis of various scenarios in which RAFT fails to estimate the optical flow map accurately. We thank the reviewer for prompting this analysis, which has in turn opened several new avenues of research for us.
>
> There are three key conclusions of this new analysis:
>
> First, in its smallest version, RAFT is very sensitive to the noise introduced by ray-tracing during rendering. This in turn makes estimation of the angle of the optical flow vector noisy, particularly in respect of large flat surfaces such as walls and ceilings.
>
> Second, RAFT has difficulties in estimating the magnitude of the optical flow vector close to the edges of the images. This is mainly because the pixel angular density of the image is higher, leading to more significant apparent motions for a slight displacement of the camera. As described in Section 4, ventral optical flow is of utmost importance for self-motion estimation in flight over the ground. The network’s failure to estimate the optical flow accurately in this location therefore presents a barrier to accurate self-motion estimation. Similarly, the network has difficulties in correctly estimating the optical flow on objects close to the camera, for which the apparent motion is fast and for which the effects of image distortion are large.
>
> Third, RAFT fails to estimate the optical flow vector's angle accurately when the apparent motion is small. As the angular density of the pixels is smallest in the central part of the image, apparent motion can be very small in this region, causing the RAFT network to fail dramatically in this specific scenario. As a result, because the RAFT was pre-trained on data largely without distortions, and with a reasonably homogeneous pixel angular density in the entire image, we see it failing in the more general scenarios encountered throughout the FlyView dataset.
>
> (2)	We also agree with the reviewer’s comment that real data could be interesting for showing the benefits of training on this new dataset. We did, in fact, collect real data in the lab with a stereo vision system composed of fisheye cameras and submillimetre tracking of the camera system using the same MoCap system presented in this paper. These real data were collected using two 8 mm focal length lenses that output images different to the equirectangular projection used in our synthetic data. The real cameras were calibrated using the Scaramuzza camera model that was then integrated into a custom-built version of Blender to reproduce the camera model with the ray-tracing system of the Blender's Cycles engine. Whilst the camera model was integrated with CUDA and OptiX enabled kernels, the resulting ground truth generation was slow and potentially inaccurate, especially around the optical axis where a strong flow singularity was observed. The singularity of the optical flow could be attenuated by augmenting the order of the Scaramuzza polynomial function used to reproject the world to the sensor, but not without having a severe impact on the computational time.
>
> We concur with the reviewer that intermediate optical flow ground truth is not required to evaluate the downstream tasks. However, training a network, understanding its internal behaviour, and quantifying the errors at each individual stage requires an accurate ground truth, which is what led us to switch from natural to synthetic images. Nonetheless, we completely agree that our initial real data collection could be of value to the reader, allowing validation of a model trained on large field-of-view images to estimate downstream tasks. We have therefore decided to release this real data alongside the FlyView dataset, together with MoCap truth data on the self-motions it contains. More information is given in the supplementary materials.

---

> ### Author Response · Authors · 2022-08-22
> **Response part 2**
>
> ## Documentation
>
> We agree with the reviewer’s comment that an explanation of how to reuse/edit the existing data pipeline should be provided. We have therefore updated our Github repository describing this process. In addition, the repository has been updated to include more details in the README file.
>
> ## Maintenance plan
>
> We thank the reviewer for commenting on our data maintenance plan. The reviewer is correct that the dataset is not fully available yet, and that only a sample version is currently available. The data will be uploaded before the conference as described in the [Dataset Embargo] section of the submission.
>
> ## Data format
>
> We thank the reviewer for their detailed comments on data format. The .blend file can be opened with Blender 2.93 LTS, and the EXR and PNG files can be opened with open source software. Our Github repository contains Python scripts to read the content from the EXR files.
> We agree that using three .blend files is suboptimal for data generation, as the scene has to be processed three times (once for each camera). Unfortunately, we encountered technical issues with Blender when attempting to collect motion flow data from multiple cameras simultaneously. To avoid these problems, we therefore decided to split our camera system into three .blend files. Each of our environments was processed in the same way. We are not aware of whether this issue has been corrected in Blender 3.0+, but as the project was initiated with Blender 2.93 LTS, we kept this version to avoid any conflict with the current code developed.
> The reviewer is correct to note the missing .csv trajectory file in our shared sample, and we thank them for spotting this. We have therefore uploaded an updated version of the sample scene containing the corresponding trajectory file.
>
> ## Additional feedback:
>
> We thank the reviewer for their additional feedback. The reviewer is correct to say that a Drosophila brain contains approximately 100,000 neurons, though comparing this to an artificial neural network remains challenging owing to the non-equivalence of the units. Nevertheless, whilst it is true to say that software is effectively without mass, no software can be functional without the hardware behind it. This hardware has a mass and a volume, which can be critical for specific applications, especially for flying vehicles with a limited payload. It is evident that the mass of a fly’s brain is much below the mass of any chip capable of running any existing artificial neural network. We therefore consider that the weight of the fly’s brain and the number of its neurons are independently important pieces of information. We have therefore opted to keep the former in the abstract, and have revised the introduction to include the latter.
>
> We also agree with the reviewer that the wording "exceptionally large" is ambiguous, even noting that the size of the dataset is close to 4Tb. We appreciate the reviewer's suggestion, and agree that whilst diverse, the number of scenes is small in comparison to the number of scenes present in some other optical flow truth datasets (Table 1). We have therefore rephrased these sentences to read:
> "FlyView is the second largest optical flow dataset in terms of the number of instances it contains, comprising 42,475 frames from each of three virtual cameras, in 9 new and diverse indoor and outdoor environments."
>
> We do agree that using the Eevee rendering engine would lead to faster data generation, but decided to use Cycles as the only engine in Blender 2.93 offering the equirectangular field of view that we used. In addition, Cycles is a physically-based rendering engine, which gave us confidence in generating photorealistic images, in particular complex illumination. Using Cycles led to a slower data generation and minor noise artefacts left by the ray-tracing in the generated visual images. Whilst we intended to minimize this noise in the scene by augmenting the number of rays used by the Cycles engine, this phenomenon cannot be entirely removed. In this sense, the data generated by Cycles mimics the output of a camera sensor introducing noise due to the ISO gain applied on the signal.
>
> We thank the reviewer for their careful and insightful review, and hope that our detailed responses to the reviewer’s comments, together with the substantive edits and additions that we have made to our revision, will indeed enable the reviewer to upgrade their assessment of our manuscript as they suggest.

---

> > ### Comment · Reviewer_6218 · 2022-08-22
> > **Missing answer**
> >
> > I'd like to thanks the authors for addressing most of my comments. I will check carefully and update my rating accordingly.
> >
> > But in the meantime, I noticed that the authors did not address the point I raised about the motivation for simulated data in Correctness. I would appreciate if the authors could provide an answer to my comment before the end of the discussion phase.

---

> > > ### Author Response · Authors · 2022-08-22
> > > **Missing response**
> > >
> > > We apologize to the reviewer. A part of the response was forgotten in the copy and paste process to add an official comment on OpenReview.
> > > Please find below the answer to the [Correctness] section of the reviewer's comment.
> > >
> > > ## Correctness
> > >
> > > We do agree with the reviewer that the approach used in the Kitti dataset is sound and valid in a context in which subpixel accurate optical flow is not required. Yet, we would like to emphasize that some applications involving ego-motion do require subpixel accuracy. For example, critical computer-vision-based surgery systems use subpixel accuracy for relative-pose estimation and ego-motion [1, 2]. Thanks to its subpixel accuracy, FlyView can be used for a larger spectrum of applications that might not have been considered by the authors yet. In addition, as answered in [Weaknesses - (2)], we now provide some real data for testing purposes. The data contains both 4K visual images of a stereo pair camera system tracked by the MoCap with a submillimetre accuracy.
> > >
> > > As discussed in the original paper at line 273-275, we purposely aimed to keep the scenes static to simplify the initial study of navigation. Dynamic objects and environments could indeed be included in an extended version of FlyView, but it is already a valuable dataset in its current form.
> > >
> > > [1] Meza, J., Romero, L. A., and Marrugo, A. G., “MarkerPose: Robust Real-time Planar Target Tracking for Accurate Stereo Pose Estimation”, 2021.
> > >
> > > [2] Michela Goffredo, Maurizio Schmid, Silvia Conforto, Tommaso D’Alessio, A markerless sub-pixel motion estimation technique to reconstruct kinematics and estimate the centre of mass in posturography, Medical Engineering & Physics, Volume 28, Issue 7, 2006,

---

### Official Review · Reviewer_icec · 2022-07-10
**A new optical flow truth dataset that contains panoramic RGB images rendered in both monocular and binocular views**

**Rating:** 8
**Confidence:** 3
**Clarity:** Yes

**Strengths:**

1- The contributions of this study are clear.
2- Attempt at leveraging large-scale, stereo and complementary sources of information.

**Weaknesses:**

(1) Page 1, Lines 1 and 17: “complex natural and indoor environments” is conceptually unclear. Shadows and occluded areas, vegetation covers, dense areas, and large-scale variation are some challenges in natural and indoor environments. Please explain this fact.
(2) Experimental benchmarking analysis (numerical and graphical) should be added to learn about the accuracy of the proposed dataset. Experimental benchmarking should provide the readers with sufficient detail about the related methods to be able to reproduce the study if so desired. The authors should be aware that in any case, benchmarking studies are normal practice when proposing a new dataset.

**Additional Feedback:**

Due to the proposed dataset being very interesting for aerospace and remote sensing researchers, a demo code with a tutorial video should be posted on github for readers.

**Correctness:**

There is no description of numerical results on state-of-the-art methods. The authors should be evaluating their dataset using several different comparisons.

**Documentation:**

The authors mention that the dataset will be fully released on Zenodo and HuggingFace, but do not provide access to the original dataset.

**Ethics:**

No ethics issues.

**Relation To Prior Work:**

Recently, a new optical flow dataset, called SAMA-VTOL, from drone images was done by [1] in 2021. The related works and Table 1 can benefit from SAMA-VTOL.
[1] https://www.int-arch-photogramm-remote-sens-spatial-inf-sci.net/XLIV-M-3-2021/1/2021/

**Summary And Contributions:**

The authors present an efficient optical flow truth dataset derived from panoramic stereo images to visual navigation. This paper attempts to highlight the capability of the proposed dataset, such as camera rendering method.

---

> ### Author Response · Authors · 2022-08-22
> **Response to reviewer**
>
> We thank the reviewer for their time taken in reviewing our work, and for their valuable comments which have helped improve the paper. We have addressed all of their comments as explained in our response below.
>
> ## Weaknesses
>
> (1) We agree that the phrase “complex environments” is rather non-specific, as we have not defined explicitly how the “complexity” of an environment is to be measured. In light of the reviewer’s suggestion, we have modified line 21 to read:
> “Changes in illumination (e.g. shadows, reflections), occlusion (e.g. vegetation, structures), and other sources of large-scale variation (e.g. different substrates, water bodies) present critical challenges to visual navigation in natural and indoor environments. Optical flow is an important visual cue that insects such as flies use to navigate such environments with a brain containing approximately 100,000 neurons and weighing only a few milligrams.”
>
> (2) We agree with the reviewer that the initial submission was lacking any benchmark on the usage of the dataset for optical flow estimation. To tackle this concern, we have added a new evaluation section testing the performance of the state-of-the-art RAFT network when pre-trained on standard computer vision datasets (FlyingChairs, FlyingThings3D, HD1K, Kitti flow, Sintel) and tested on FlyView. This new section demonstrates that the pre-trained RAFT network performs poorly when evaluated on FlyView, showing the importance of taking account of image distortion when applying state-of-the-art optical flow learners to widefield images. In addition, we offer a qualitative analysis of various scenarios in which RAFT fails to estimate the optical flow map accurately. We thank the reviewer for prompting this analysis, which has in turn opened several new avenues of research for us.
>
> There are three key conclusions of this new analysis:
>
> First, in its smallest version, RAFT is very sensitive to the noise introduced by ray-tracing during rendering. This in turn makes estimation of the angle of the optical flow vector noisy, particularly in respect of large flat surfaces such as walls and ceilings.
>
> Second, RAFT has difficulties in estimating the magnitude of the optical flow vector close to the edges of the images. This is mainly because the pixel angular density of the image is higher, leading to more significant apparent motions for a slight displacement of the camera. As described in Section 4, ventral optical flow is of utmost importance for self-motion estimation in flight over the ground. The network’s failure to estimate the optical flow accurately in this location therefore presents a barrier to accurate self-motion estimation. Similarly, the network has difficulties in correctly estimating the optical flow on objects close to the camera, for which the apparent motion is fast and for which the effects of image distortion are large.
>
> Third, RAFT fails to estimate the optical flow vector's angle accurately when the apparent motion is small. As the angular density of the pixels is smallest in the central part of the image, apparent motion can be very small in this region, causing the RAFT network to fail dramatically in this specific scenario. As a result, because the RAFT was pre-trained on data largely without distortions, and with a reasonably homogeneous pixel angular density in the entire image, we see it failing in the more general scenarios encountered throughout the FlyView dataset.
>
> ## Relation to Prior Work:
>
> We thank the reviewer for bringing this reference to our attention, which provides interesting annotations, including semantic and panoptic segmentation, albeit without the large field-of-view images and optical flow ground truth that FlyView provides  . Because Table 1 only lists datasets that include optical flow ground truth, we have not included SAMA-VTOL in Table 1, but have instead cited it in the main text as an excellent example of current trends in UAV technology developments,  and as an important dataset for UAV navigation in general.
>
> ## Additional Feedback
>
> We thank the reviewer for their additional feedback and agree that the dataset would benefit from a tutorial video. We plan to include tutorial videos for (a) using FlyView for model training and (b) changing the camera model in the sample scene for reproducibility. We plan to have these videos shared with the community and released at the NeurIPS conference. In the meantime, we have our Github repository explaining how to use the existing sample to (i) reproduce FlyView, (ii) extend FlyView, and (iii) create new data using a similar pipeline with custom camera parameters.

---

### Official Review · Reviewer_3Tcv · 2022-07-21
**FlyView is a novel and valuable work for visual navigation.**

**Rating:** 7
**Confidence:** 4

**Strengths:**

significance of the contribution:
1.FlyView builds a bio-inspired optical flow truth dataset, that mimics insects to navigate by using optical flow with compound eyes.
2.FlyView includes dynamically meaningful self-motion, and is the first optical flow truth dataset for motion flow and self-motion estimation.

relevance to the broader research community:
1.FlyView offers an almost complete field of view for visual images, motion flow, and depth maps in diverse indoor and outdoor environments, and these data have rich research value for many tasks such as optical flow estimation, depth estimation, and self-motion estimation.
2.FlyView includes special panoramic RGB images modelled on the visual field of blowflies' compound eyes, which is of bionic significance and brings new challenges to both biology and computer vision.

accessibility and accountability:
The dataset will be uploaded to Zenodo and HuggingFace, and download links will be shared on the Github repository.

ethical and social implications:
1.The dataset is self-contained and does not rely on any other dataset.
2.The dataset does not contain data that might be considered confidential
3.The dataset does not contain data that might be offensive, insulting, threatening, or might
cause anxiety, and is not related to people.

**Weaknesses:**

1.The dataset is not currently implemented on a specific task or lacks a baseline model.

2.The font in the figure 5-7 is too small.

3.Appendix B can give more explanations about motion flow and scene frames.

**Additional Feedback:**

First, although the dataset can be used to optical flow estimation, depth estimation, and self-motion estimation, the paper could discuss how the dataset are implemented on these tasks, especially in self-motion estimation.

Second, there are few experiments in this paper, which prove the rationality and availability of the dataset or prove the advantages of bionic optical flow dataset.

**Clarity:**

The paper basically describes the the motivation and methods of dataset establishment. The theoretical starting-point of this paper is logical. The writing quality is good.

**Correctness:**

The dataset was generated using Blender 2.93 LTS and its cycle engine for ray-tracing. Because visual images were generated using ray-tracing in Blender, the pixel intensity generated have to be post-processed to remove potential noise.

**Documentation:**

Yes.

**Ethics:**

No.

**Relation To Prior Work:**

Yes, the dataset has several key features that are missing from existing optical flow datasets, including:
1. panoramic cameras with a monocular and binocular field of view matched to that of a fly’s compound eyes;
2. dynamically meaningful self-motion modelled on motion primitives, or the 3D trajectories of drones and flies;
3. complex natural and indoor environments including reflective surfaces.

**Summary And Contributions:**

This work presents a novel optical flow dataset with panoramic stereo images to estimate self-motion in flight. The authors reveal the importance of downstream tasks (such as self-motion estimation) for insects' navigation, so this bio-inspired optical flow dataset is designed to supplement the lack of such research in academia.

The main contributions of this paper are:
1.FlyView includes special panoramic RGB images modelled on the visual field of blowfly, which is of bionic significance and brings new challenges to both biology and computer vision.
2.FlyView is the first optical flow truth dataset to include motion dynamics relevant to flight.
3.FlyView offers an almost complete field of view for visual images, motion flow, and depth maps in diverse indoor and outdoor environments, and these data have rich research value for visual navigation.

---

> ### Author Response · Authors · 2022-08-22
> **Response from authors**
>
> We thank the reviewer for their time taken in reviewing our work, and for their insightful comments which have helped improve the paper. We have addressed all of their comments as explained in our response below.
>
> ## Weaknesses:
>
> We thank the reviewer for pointing out that our paper would be strengthened by testing the dataset against a baseline model. To tackle this concern, we have added a new evaluation section testing the performance of the state-of-the-art RAFT network when pre-trained on standard computer vision datasets (FlyingChairs, FlyingThings3D, HD1K, Kitti flow, Sintel) and tested on FlyView. This new section demonstrates that the pre-trained RAFT network performs poorly when evaluated on FlyView, showing the importance of taking account of image distortion when applying state-of-the-art optical flow learners to widefield images. In addition, we offer a qualitative analysis of various scenarios in which RAFT fails to estimate the optical flow map accurately. We thank the reviewer for prompting this analysis, which has in turn opened several new avenues of research for us. Further detail on the results of this analysis may be found in our responses to Reviewers 1 and 3.
>
> We have also followed the reviewer’s suggestions to increase the font size in Figure in section B in the Supplementary Materials and added a new Section B3 in Supplementary Materials to include 3D tracks of the trajectories performed in FlyView.
>
> ## Correctness
>
> The reviewer is right to note that noise can be created when using the Cycles engine from Blender 2.93LTS, owing to the ray-tracing approach used to generate the visual images. Whilst the presence of a large number of rays can reduce the uncertainty on a specific pixel, we cannot thereby ensure that the brightness of a pixel remains constant between two successive images. This is similar to the behaviour of a real camera sensor that includes noise in the 2D signal. In particular, this phenomenon is clearly visible on dark scenes in which the camera sensor usually applies a high ISO gain to correct brightness in the scene. The higher the ISO gain applied, the higher the noise in the signal.
>
> ## Additional Feedback
>
> We agree that the applications and implementations of the dataset were missing and have included our supplementary materials detailing this in response to the other reviewers. In addition, as answered in the [Weaknesses] section, we have added a new section comparing the performances of a pre-trained RAFT on several datasets, including FlyView, and have highlighted scenarios in which RAFT fails to estimate the optical flow correctly.

---

### Official Review · Reviewer_yzAj · 2022-07-27
**This paper provides a panoramic view dataset, inspired from how biological insects view a scene. This dataset provides optical flow, self-motion, and RGB view ground truth data.**

**Rating:** 6
**Confidence:** 4

**Strengths:**


**Large-scale dataset**:

The major strength of this work is the generation of a simulated large scale optical flow, camera trajectory, RGB, and depth ground dataset with panoramic views. The camera trajectories in this dataset are dynamically meaningful since they mimic the trajectories of insects and drones.

**Generalization ability**:

Moreover, since this dataset is covering varied environments, one strength of this approach is that the authors have taken into consideration what data splits are relevant keeping the generalization ability in mind if this dataset is used to train deep neural networks.

**Aerial vehicle research**:

Since the camera trajectories mimic the trajectories of drones and insects, this dataset could find usage in the aerial vehicle community where they could train optical flow estimation or self-motion estimation tasks in simulation and replicate them in the real world.


**Weaknesses:**

**Quantification of improvement**:

One of the areas where this submission could be improved is in the task of showing how much value does this dataset actually bring to the table. The authors did not show any results on how much improvement can this dataset bring in the tasks of optical flow or self-motion estimation compared to earlier datasets. One simple way to do this would have been to pick any state-of-the-art (SOTA) methods for optical flow estimation or self-motion estimation, run them on existing datasets, and compare the same SOTA method using the proposed dataset. This metric could be very useful for the reader to understand where this dataset is bringing value or providing something which was not available in the earlier datasets.


**Related works**:

Although it is understandable that the authors are trying to formulate the problem as a panoramic view dataset mimicking the flying vehicles or flying insects, it would be useful for readers if the authors add the following related works, for completeness.
1. Yogamani, Senthil, et al. "Woodscape: A multi-task, multi-camera fisheye dataset for autonomous driving." Proceedings of the IEEE/CVF International Conference on Computer Vision. 2019.
2. Artizzu, Charles-Olivier, et al. "Omniflownet: a perspective neural network adaptation for optical flow estimation in omnidirectional images." 2020 25th International Conference on Pattern Recognition (ICPR). IEEE, 2021.
3. M. Yuan and C. Richardt, “360° optical flow using tangent images,” in Proc. BMVC, 2021
4. A. R. Sekkat et al., “SynWoodScape: Synthetic surround-view fisheye camera dataset for autonomous driving,” IEEE Robotics and Automation Letters, 2022.


**Additional Feedback:**

It would be great to know what the authors respond to the questions stated in **Weaknesses** section.


**Also, a question to the authors concerning drone research:**

How well do you think a drone trained on the simulated dataset would perform in the real world? Domain adaptation from sim-to-real has been an area of interest by many drone researchers and I wonder if this dataset could bridge that gap. Since many deep neural networks are running on the i.i.d. assumption, many times the domain change from simulation to real brings with itself the complicated problem of out-of-distribution samples. How well do you think the simulated data work in such scenarios? Are there any good precedents in the drone research where a system trained on simulated data transfers the knowledge reasonably well in the real world data?

**Clarity:**

The paper is well written and easy to follow. There are no major issues with the way the dataset is documented and the datasheet submitted along with the dataset covers all the details relevant to the usage of the dataset.

**Correctness:**

The claims made in the submission are accurate and the dataset is constructed in a sound way. The datasheet is also provided along with the dataset in the supplementary materials which further covers the details of the dataset.

**Documentation:**

Since the datasheet is provided with the original dataset, sufficient detail on data collection, organization, maintenance, ethical, and responsible is easily understandable by the reader. Moreover, the authors also provide a URL to access a sample dataset and the authors promise to release the full dataset on Zenodo and HuggingFace -- thereby answering the questions concerning hosting, licensing, and maintenance of the dataset.

**Ethics:**

The datasheet associated with the dataset reasonably answers the ethical concerns of the collected dataset.

**Relation To Prior Work:**

Please refer to the **Weaknesses** section.

**Summary And Contributions:**

**Summary of contribution**:
- This paper is generating a dataset that could be used for training deep learning models of optical flow estimation for self-motion estimation.
- The paper mentions that there does not exist a dataset with panoramic view that provides optical flow ground truth for estimating "self-motion" -- where the "motion" is the flying pattern of an insect or a drone (micro-air vehicle).
- To this end, they propose a dataset with panoramic views that mimics how insects and drones fly in different environments.
- Quantitative improvement benchmarking on optical flow estimation or self-motion estimation tasks using this dataset is missing.


**Dataset information**:
- They generate the **synthetic data** using Blender and place the **three camera views** to cover **340 degree views** of the environment.
- They also provide intrinsics to convert between different camera views based on the application. Although authors mention this, I would like to know more about this claim -- for example: it would be really useful for the community if the authors could provide a streamline way of converting from panoramic view camera model to the generic pinhole camera model so that it could be used in different applications.
- RGB images, Forward-Backward motion flow, Self-motion (trajectory of the cameras), and Depth Map are provided for all the views.
- Dynamically meaningful camera trajectories are provided which could mimic how the insects and drones fly in the real world.
- Some comparative datasets are missing in the related works.

---

> ### Author Response · Authors · 2022-08-22
> **Response authors part 1**
>
> We thank the reviewer for their time taken in reviewing our work, and for their insightful comments which have helped improve the paper. We have addressed all of their comments as explained in our response below.
>
> ## Summary and contributions
> ### Summary and contribution
>
> We agree with the reviewer that the paper did not include benchmarking on optical flow estimation or self-motion estimation using our dataset. Our revised paper now includes a section dedicated to the performance of existing state-of-the-art models on our dataset and others.
>
> ### Dataset information
>
> Regarding camera parameters, we currently provide the intrinsic and extrinsic parameters of the three cameras from which the camera system is composed. The reviewer's question focuses on converting the intrinsic parameters of a large field-of-view camera model to a pinhole camera model. Currently, the intrinsics are specified according to the Scaramuzza camera model [1]. To the best of our knowledge, there is no direct way to convert this model to a pinhole camera model analytically. We do, however, provide images of calibration boards, allowing the user to compute the intrinsics using a different camera model should they wish to. Due to the very large field of view, it is unlikely that a pinhole camera model will suffice to describe the data. Nevertheless, the user can use different camera models, such as those by Mei [2] and Kannala-Brandt [3], which are already implemented in OpenCV and OpenCV-contrib.  We have cited this reference accordingly in our revised paper and supplementary materials.
>
> We have responded to the reviewer's comment that some comparative datasets are missing in the related works section in our response to the specific comment [Weaknesses – Related Works] below.

---

> ### Author Response · Authors · 2022-08-22
> **Response part 2**
>
> ## Weaknesses
>
> We thank the reviewer for their insightful comment. In order to quantify the benefits of our FlyView dataset, we have added a new evaluation section testing the performance of the state-of-the-art RAFT network when pre-trained on standard computer vision datasets (FlyingChairs, FlyingThings3D, HD1K, Kitti flow, Sintel) and tested on FlyView. This new section demonstrates that the pre-trained RAFT network performs poorly when evaluated on FlyView, showing the importance of taking account of image distortion when applying state-of-the-art optical flow learners to widefield images. In addition, we offer a qualitative analysis of various scenarios in which RAFT fails to estimate the optical flow map accurately. We thank the reviewer for prompting this analysis, which has in turn opened several new avenues of research for us.
>
> There are three key conclusions of this new analysis:
>
> First, in its smallest version, RAFT is very sensitive to the noise introduced by ray-tracing during rendering. This in turn makes estimation of the angle of the optical flow vector noisy, particularly in respect of large flat surfaces such as walls and ceilings.
>
> Second, RAFT has difficulties in estimating the magnitude of the optical flow vector close to the edges of the images. This is mainly because the pixel angular density of the image is higher, leading to more significant apparent motions for a slight displacement of the camera. As described in Section 4, ventral optical flow is of utmost importance for self-motion estimation in flight over the ground. The network’s failure to estimate the optical flow accurately in this location therefore presents a barrier to accurate self-motion estimation. Similarly, the network has difficulties in correctly estimating the optical flow on objects close to the camera, for which the apparent motion is fast and for which the effects of image distortion are large.
>
> Third, RAFT fails to estimate the optical flow vector's angle accurately when the apparent motion is small. As the angular density of the pixels is smallest in the central part of the image, apparent motion can be very small in this region, causing the RAFT network to fail dramatically in this specific scenario. As a result, because the RAFT was pre-trained on data largely without distortions, and with a reasonably homogeneous pixel angular density in the entire image, we see it failing in the more general scenarios encountered throughout the FlyView dataset.
>
> ## Related works
>
> We thank the reviewer for providing this interesting list of references. We were unaware of these datasets and are happy to see more developments on large field-of-view images for optical flow. We have added these references to the related works section for completeness.

---

> ### Author Response · Authors · 2022-08-22
> **Response part 3**
>
>
> ## Additional feedback
>
> We thank the reviewer for their interesting question regarding how well we think that a network trained on simulated data will transfer to real data from drones. We agree with the reviewer that domain adaptation from sim-to-real is challenging and far from perfectly achieved. For example, because the small RAFT network was trained on simple and "perfect" synthetic data, the network is very sensitive to the noise arising from ray-tracing in FlyView. We can expect the same behaviour for such a pre-trained network that would be fed by natural images containing motion blur, ISO gain, lens aberrations, or other artefacts from the camera's rolling shutter. We agree that the I.I.D assumption has its limitations when working with synthetic data. Yet, synthetic data remain essential to validate numerous components and blocks of the algorithms.
>
> To reduce the risk of a deep network performing poorly after training on simulated data, we see some exciting approaches emerging in the market, such as GAN networks being used to make synthetic data look "more realistic". We believe that this approach is promising. Not only can GAN be used to enhance simulated data, but also to avoid out-of-distribution samples on natural data. Indeed, seasonal change is a great challenge for drone navigation as features such as vegetation will change completely and GAN can be part of the solution to tackle this issue [4, 5]. Another possibility is to use real data collected in a motion capture setup like the one we describe, but a disadvantage of that approach is the difficulty in collecting comparably accurate truth data on the scene. We would therefore foresee this real-world data as being most useful in validating the accuracy of self-motion estimation in real world scenarios for an optical flow learner trained on simulated data.
>
> Finally, one great example of transfer from sim-to-real is given in [6], where the authors proposed an abstraction-based strategy to transfer a policy trained in simulation to a physical vehicle. They train a control policy from abstract representations (e.g. semantic segmentation) instead of the raw sensory inputs (e.g. color images). This training strategy enormously simplifies the transfer problem since the abstract representations are more similar between domains than the raw sensor measurements.
>
> [1] Scaramuzza and al. 2006, A Flexible Technique for Accurate Omnidirectional Camera Calibration and Structure from Motion, Conference: 2006 IEEE International Conference on Computer Vision Systems, January 5-7, 2006, St. Johns University, Manhattan, New York City, New York, NY, USA, Proceedings, CDROM
>
> [2] Mei and al. 2007, Single view point omnidirectional camera calibration from planar grids, in ICRA 2007.
>
> [3] Kannala and al. 2006, A Generic Camera Model and Calibration Method for Conventional, Wide-Angle, and Fish-Eye Lenses, IEEE Transactions on Pattern Analysis and Machine Intelligence 28(8):1335-40
>
> [4] J. Zhu, T. Park, P. Isola and A. A. Efros, "Unpaired Image-to-Image Translation Using Cycle-Consistent Adversarial Networks," 2017 IEEE International Conference on Computer Vision (ICCV), 2017, pp. 2242-2251, doi: 10.1109/ICCV.2017.244.
>
> [5] Paudel and al. “Landscape Image Season Transfer Using Generative Adversarial Networks”, Proceedings of 10th IOE Graduate Conference, 2021
>
> [6] Matthias Müller et al. “Driving Policy Transfer via Modularity and Abstraction”. In: Conference on Robot Learning. 2018, pp. 1–15.

---

### Official Review · Reviewer_EU1v · 2022-07-27
**A pipeline to render photo-realistic  using Blender, inspired by the blowfly**

**Rating:** 6
**Confidence:** 3
**Correctness:** 1) Please use the correct taxonomic i…

**Strengths:**

1) Replicable code and creating of high fidelity (photo-realistic) images using Blender
2)  Application of the pipeline to render images with different motion attributes and FOV.
3) Good description of the Flyview dataset
4) The use of optical flow



**Weaknesses:**

1) Unclear which annotations is provided in the dataset, reading the manuscript. This is clarified in the Suppl. Information, but should be clearly stated in the manuscript.
2) The manuscript is centered around a set of images (The FlyView dataset) which might not be generalized to many applications.
3) The authors should bonify section 2 and 4.1 with specific application in biological vision and use-case for the dataset.
4) The rendering pipeline including the inputs and outputs should be described in more details.
5) There is, beside the depth information, no annotation of object in the dataset (bounding boxes or other), limiting its application in various Machine Learning training application.
6) Unclear if some environmental conditions (e.g. weather, flare) are included. This is important since in real-world applications, cloud covers or other factors will change the illumination of surfaces.
7) While training of novel computer vision algorithms and benchmark is mentioned in the conclusion, a Table of compatible drone or robotic platform could be provided to enhance the usefulness of this dataset in computer vision applications.


**Additional Feedback:**

The ideas presented in this manuscript are interesting. However, the dataset per se seems limited in the research application and use of the dataset. A better overview of the use-case scenario for this dataset might be provided by the authors.

Otherwise, a schematic view of the rendering process including input/output and rendering time per frame could be provided in the main manuscript. There is clearly a need for similar datasets in agriculture and automated driving or pipeline for such images generation. However, since different camera placement, type and characteristics will change the FOV and the resulting images, and since training DL networks required some fixed parameters, this part of the manuscript should be improved.


**Clarity:**

The authors described the dataset and provided a datasheet. The text is well written, however, some information (see above) would complement this manuscript.

**Documentation:**

No problem.

**Ethics:**

No problem.

**Relation To Prior Work:**

Section 3.1 give a good description of the dataset with Table 1 and section 2.

**Summary And Contributions:**

In this manuscript, Leroy and Taylor describe both a generator of scenes using Blender, suitable to create pictures to train drones and robots for navigation using machine learning, and a dataset of 42,475 images containing depth information. Their dataset, FlyView, is calibrated to the FOV of the blowfly Calliphora stygia. The generator is impressive, allowing the use of custom trajectory with rotation and translation. One drawback of the current dataset is the low picture quality (1700x900 px), with new cameras having >4K resolution and the annotation of the different images in terms of object types (class/category), locations (3d space) and composition (environmental effects such as weather?) that is not presented in the manuscript. Also, the section comparing this dataset to the similar datasets  (section 4.2) could be improved. Nevertheless, the manuscript and developed Blender pipeline could be of in different training of neural networks, given that some information could be provided by the authors in modifying their pipeline to accommodate for other camera types and location.

---

> ### Author Response · Authors · 2022-08-22
> **Response part 1**
>
> We thank the reviewer for their time taken in reviewing our work, and for their valuable comments which have helped improve the paper. We have addressed all of their comments as explained in our response below.
>
> ## Summary and Contribution:
>
> The reviewer is correct to state that modern camera systems enable high-resolution images at 4K resolution or higher, but in the context of optical flow estimation, an image of 1700x900 px is by no means low resolution. On the contrary, as Table 1 shows, FlyView (42,475 frames at 1700x900 px) has the highest image resolution of any optical flow truth dataset apart from HD1K (which contains only 1137 frames at 2560x1080 pixels). Moreover, even a standard NVIDIA 3090 GPU with 24Gb of RAM has insufficient GPU memory to perform inference for the state-of-the-art network RAFT on a FlyView panoramic view instance. Were we to have provided higher resolution, the user would be left with the options of: (i) reducing the input image resolution to the GPU; (ii) using more powerful, but expensive, GPUs; or (iii) using a smaller network that does not achieve state-of-the-art results. In addition, training such a network through backpropagation would require even more compute.
> The reviewer is correct to say that we include no object annotation or locations in the scenes contained in FlyView. This simply reflects our focus on optical flow estimation and self-motion estimation, and it would of course be possible for another user to add such information if desired with reference to the 3D assets. On the other hand, we do include fog, water reflections, and clouds in the visual images. As discussed in the first version of the paper at line 273-275, we purposely aimed to keep the scenes static to simplify the initial study of navigation. Dynamic objects and environments could indeed be included in an extended version of FlyView, but it is already a valuable dataset in its current form.
> The reviewer also pointed out that Section 4.2 could be improved to compare FlyView with existing datasets . We developed this section by adding the following:
>
> “Additionally, many standard datasets lack well-defined validation and test sets. Some provide visual images without motion flow ground truth (e.g. MPI Sintel), others do not provide any defined test set (e.g. Monkaa and Driving), or any defined validation set (e.g. FlyingThings3D, FlyingChairs). In contrast, FlyView has been designed to include both a validation and a test set with accurate ground truth. Appendix C provides a detailed description of our recommended split.”
> Finally, whilst the reader can download the sample scene from the Github repository and explore the example project to construct a similar data generation pipeline, we agree with the reviewer that a description of the key steps to achieve this would improve the overall project. Given space constraints, we have therefore added a brief description of these key steps to GitHub repository.

---

> ### Author Response · Authors · 2022-08-22
> **Response part 2**
>
> ## Specific comments:
>
> 1. In response to the reviewer’s comment, we have expanded Section 1 to summarise the scenes and their content, cross-referencing the more detailed description of the content of the 3D assets in Supplementary Material.
>
> 2. Please see our response to the reviewer’s general comments above, which addresses this point in detail.
>
> 3. We have followed the reviewer’s suggestion to add explanations of specific applications in biological vision and potential use-cases in Sections 2 and 4.1 as follows:
>
> At line 112-113, we have included the following :
>
> “Apart from the limitations inherent to existing computer vision datasets, no analogous dataset could be found containing input luminance, motion flow, motion state, and depth data for a flying insect. Such data could enable important breakthroughs in modelling the visual system of insects, including modelling the response of specific neurons, and offering better understanding of their mechanisms of optical flow computation. For these reasons, we decided to design and collect our own synthetic dataset matching bio-inspired self-motion estimation requirements.”
>
> We agree with the reviewer that Section 4 lacks some information on potential applications, and have added these to a new Section G in the Supplementary Materials that also takes account of suggestions made by other reviewers.
>
> 4. In response to the reviewer’s comment, we have updated the GitHub repository to include a description of the pipeline’s expected inputs and outputs.
>
> 5. Please see our response to the reviewer’s general comments above, which addresses this point in detail.
>
> 6. We agree with the reviewer that environmental changes and other factors will impact the scene, and used ray tracing to model various environmental conditions including clouds, fog, and reflections. All of these will affect the scene differently according to where the agent is located. See also our response to the reviewer’s general comments above.
>
> 7. We thank the reviewer for their suggestion to include a comment on potential platforms that could make us of such a dataset. The new section G discussing possible applications of our dataset now includes some examples of this.
>
> ## Correctness
>
> 1. We have now used the full scientific names “Drosophila melanogaster” and “Calliphora vicina” at their first occurrence, but refer to these thereafter as Drosophila and Calliphora, in accordance with biological convention for these model species.
>
> 2. Figures A.1, B.1 and B.2  provide summary information that readers may find useful in assessing possible biases present in the dataset, but we agree that a 3D figure showing representative tracks would be helpful, and have added this accordingly in Section B3 in the Supplementary Materials.
>
> 3. We agree that no metrics were given comparing real and simulated datasets, but training of a machine learning model on our dataset is required before evaluating its performance on real data. Our research activities include collecting real images using large field-of-view cameras, which as stated in our conclusion will form part of our future work.
>
> ## Additional Feedback
>
> We thank the reviewer for their constructive comments, which have helped us to improve our manuscript, and have responded to their additional feedback in answering the queries raised above.

---

### Official Review · Reviewer_6z2S · 2022-07-28
**Interesting work with room for improvement**

**Rating:** 6
**Confidence:** 3

**Strengths:**

I really like the general direction of this paper, which provides some insights about how insects fly in complex environments. I indeed appreciate the deep knowledge of authors in the bio field, and how they tried to incorporate this knowledge in a virtual environment. In general, I am interested in combining the knowledge in various fields, and I believe authors’ general research direction will be really fruitful. I completely agree with the authors that using optical flow and wide field of view cameras can indeed improve navigation systems. Moreover, the dataset acquisition procedure is well described in detail.

**Weaknesses:**

While I appreciate the authors’ efforts and their deep knowledge in the bio field, I have some suggestions/concerns related to the vision field that might help improve the work, as described in the following.

**1. [Self-contained data?]:** In the supplementary (page 2), it is written that the dataset is self-contained. However, in the github page, only a sample of data is provided, which contains an example trajectory in a simple scene. As mentioned in the supplementary (page 5), authors do not own the rights to the 3D assets they used. Therefore, it seems that for each virtual scene, the code should be used to render and generate the data. Based on the paper (line 182-184), rendering time might take over a period of a few months. Authors should clarify whether they will provide the final rendered data or future users must render the scenes to generate the data. Without providing the final data, it will be really hard to use the dataset.

**2. [No Experiments]:** No experiments or evaluations are provided to support that this dataset can really improve the claimed applications in the computer vision domain: optical flow estimation, depth estimation, and self-motion estimation. Authors mentioned that this dataset is deliberately designed to help avoid over-fitting but no evidence is provided. In particular, the total number of virtual scenes used to create this dataset is 9 (2 outdoor and 7 indoor scenes), and I am not sure this is sufficient to generalize well on unseen real world scenes. To clarify the usefulness of this dataset, a comparison between 1) a network (e.g,, an optical flow estimator) trained using this dataset with 2) a network trained using existing datasets (or combination of this data with other ones) will be really useful.

**3. [No Training]:** In the conclusion (line 339-340), authors mentioned that “Our current work uses these data to train novel bio-inspired computer vision algorithms”. In the paper, I could not find any training procedure. This paper only provides a bio-inspired dataset without training it on any existing or proposed bio-inspired network architectures. Readers will benefit if authors clarify what they mean by training novel bio-inspired computer vision algorithms.

**4. [Trajectory designs seem independent to the environment]:** What makes insects interesting is their interaction with complex environments and their ability to fly and navigate in complex scenes with respect to objects in their surroundings. However, it seems that the camera trajectories are designed without considering the interaction with the environment. As far as I understood, some predefined camera trajectories are rendered in different virtual environments. This dataset can be more useful, if one could use this data to imitate the insect fly navigation (path planning) with respect to objects in a complex environment.

**5. [Why dynamically meaningful self-motion?]:** Authors might clarify why using dynamical meaningful motions is more effective than using any arbitrary motions in the self-motion estimation. Why does using insect motions help improve self-motion estimation in general applications? If the final goal was path planning or imitating flying paths of insects, it will be obvious that we must generate motion data similar to those of the insect motions. However, when the final goal is the self-motion estimation, maybe even using more various camera path shapes (e.g., splines with various shapes and moving cameras on them with different velocities in the range of insect or drone physical limitations)  provides a richer dataset, which covers a wider range of possible camera motions. To prove the claim that dynamically meaningful self-motion is useful for self-motion estimation, an experiment might be needed to compare a self-motion estimator trained using arbitrary paths against the one trained on dynamically meaningful self-motion paths.

**6. [Application]:** Readers will benefit if authors can be more specific about the applications of this dataset in studying the insects vision system. Moreover, I think this data will benefit drone navigation if that drone can be as small as an insect or can have the same camera setup (2 cameras with 4 mm distance). However, I am not sure about its feasibility, considering current drone sizes and camera systems. If no drone exists with a similar camera setup, I am not sure how using 4mm stereo baseline can benefit drone self-motion estimation or path planning. Moreover, I am not sure why it is important to have 340 degree coverage similar to insects for drone navigation if having a 360 view field is possible for drones using one 360 VR camera.


**Additional Feedback:**

- In Table 2, for the blowfly Calliphora, 12,700 frames are provided with the frame per second rate of 16,620. This means that the whole data duration is less than 1 second for 36 different insect trajectories. Is this short amount of time enough for providing data that is a good representative of insect fly dynamics?

- If the purpose is self-motion estimation, drones have onboard sensor data that work quite well for estimating camera motion. What is the advantage of using this data to train a self-motion estimator over using onboard drone sensors?

- Line 339:  “featuresextraction” should be changed to “feature extraction”


**Clarity:**

It can be improved by further clarifying the applications of this dataset. In general, why this dataset should be used instead of existing ones for the claimed computer vision tasks (e.g., optical flow estimation) or how this dataset will improve the results of claimed tasks if it is aggregated with current datasets for training.

**Correctness:**

The data collection procedure seems technically correct.


**Documentation:**

Dataset is well documented.

**Ethics:**

The paper seems to have no ethical problems.


**Relation To Prior Work:**

While the paper tries to faithfully discuss the relation to prior studies, it will benefit readers if authors further clarify the differences between this dataset and the ones in Table 1 with the checked stereo feature (column 6). Moreover, there are many studies in the computer graphics literature (especially TOG/SIGGRAPH) related to aerial/drone motion path planning and drone trajectory design, which are missing. These studies might help designing better drone trajectories (used in section 3.3.2) or even using their drone dynamical models to generate synthetic drone trajectories without a need for flying in a motion capture system. These dynamical models also consider physical limitations of the drone, similar to a real experiment.

**Summary And Contributions:**

This paper provides a dataset to train AI models on computer vision tasks such as optical flow estimation, depth estimation, and self-motion estimation. The dataset is gathered by rendering camera views in virtual environments. The stereo cameras are designed to have similar attributes to insect eyes. The dataset includes synthetic visual views, motion flow, position, and orientation of the cameras. Cameras are designed to follow the visual features similar to insect eyes (4mm stereo baseline). The dataset gathering is inspired from insects but it does not provide any bio-inspired network architecture designs.

---

> ### Author Response · Authors · 2022-08-22
> **Response #1**
>
> Beginning Response 1
>
> We thank the reviewer for their time taken in reviewing our work, and for their valuable comments, which have helped improve the paper. We have addressed all of their comments as explained in our response below.
>
> # Weaknesses :
>
> ## 1. [Self-contained data?]
>
> The reviewer is correct that the github repository only contained one sample dataset at the time of manuscript submission. However, as we have now described more clearly in Section 1 and in the Supplementary Materials, the FlyView dataset is indeed complete and self-contained. The complete FlyView dataset contains 42,475 instances comprising the rendered images, depth maps, and optical flow maps for the three cameras, together with the position of the agent. Assuming that our manuscript is accepted for publication, these 42,475 instances will all be provided through an open access repository  at the point of publication. All the rendered images and truth data will be freely downloadable, allowing the reader to use the FlyView dataset for model training or evaluation without any need for further rendering.
>
> The reviewer is correct that we do not own the rights to all the 3D assets, but we only used third-party assets with a license that would enable us to share the resulting dataset, and these are not required for model training or evaluation. Instead, a link to the artist's repository is given in the Supplementary Materials, enabling the reader to download any assets needed to reproduce the analysis independently if desired. We do, of course, provide access to those 3D assets that we generated ourselves during data collection.
>
> ## 2. [No experiments]
>
> We generated FlyView to provide a novel dataset suitable for research on insect vision and complementing standard datasets in computer vision. Publication of the FlyView dataset will enable current computer vision training and evaluation pipelines to be extended with the addition a significant number of new instances (42,475 instances from 2 outdoor and 7 indoor scenes). We do, however, appreciate the reviewer’s comment that experiments or evaluations are necessary to demonstrate the usefulness of the FlyView dataset. To tackle this concern, we have added a new evaluation section testing the performance of the state-of-the-art RAFT network when pre-trained on standard computer vision datasets (FlyingChairs, FlyingThings3D, HD1K, Kitti flow, Sintel) and tested on FlyView. This new section demonstrates that the pre-trained RAFT network performs poorly when evaluated on FlyView, showing the importance of taking account of image distortion when applying state-of-the-art optical flow learners to widefield images. In addition, we offer a qualitative analysis of various scenarios in which RAFT fails to estimate the optical flow map accurately. We thank the reviewer for prompting this analysis, which has in turn opened several new avenues of research for us.
> There are three key conclusions of this new analysis:
> First, in its smallest version, RAFT is very sensitive to the noise introduced by ray-tracing during rendering. This in turn makes estimation of the angle of the optical flow vector noisy, particularly in respect of large flat surfaces such as walls and ceilings.
> Second, RAFT has difficulties in estimating the magnitude of the optical flow vector close to the edges of the images. This is mainly because the pixel angular density of the image is higher, leading to more significant apparent motions for a slight displacement of the camera. As described in Section 4, ventral optical flow is of utmost importance for self-motion estimation in flight over the ground. The network’s failure to estimate the optical flow accurately in this location, therefore, presents a barrier to accurate self-motion estimation. Similarly, the network has difficulties in correctly estimating the optical flow on objects close to the camera, for which the apparent motion is fast and for which the effects of image distortion are large.
> Third, RAFT fails to estimate the optical flow vector's angle accurately when the apparent motion is small. As the angular density of the pixels is smallest in the central part of the image, apparent motion can be very small in this region, causing the RAFT network to fail dramatically in this specific scenario. As a result, because the RAFT was pre-trained on data largely without distortions, and with a reasonably homogeneous pixel angular density in the entire image, we see it failing in the more general scenarios encountered throughout the FlyView dataset.
>
> End Response 1

---

> ### Author Response · Authors · 2022-08-22
> **Response #2**
>
> Beginning response 2
>
> ## 3. [No training]
>
> We thank the reviewer for pointing out the unintended ambiguity of this sentence, which was meant to refer to our planned and ongoing research activities. This has been modified to read: " Future work will use these data to train novel bio-inspired computer vision algorithms, and to provide a new benchmark for existing state-of-the-art algorithms.”
>
> ## 4- Trajectory designs seem independent to the environment
>
> We thank the reviewer for their very interesting comment on the close coupling between insect vision and insect navigation. As the reviewer notes, virtual cameras were used to render images and truth data at different timesteps, having located a pre-defined trajectory in a pre-defined environment. The virtual agent following trajectories collected empirically from blowflies is therefore moving through the environment with realistic flight dynamics but is not reacting to its environments. We agree that it would be highly worthwhile to generate such data, and have done so in another context for birds flying through cluttered indoor environments (see Miñano, S. & Taylor, G.K., 2021, bioRxiv https://doi.org/10.1101/2021.06.16.446415). This is a project in its own right, however, which we expect to conduct with blowflies in due course.  In the meantime, we note that whereas embedding the navigation dynamics in the training data would be important if the navigation dynamics were the focus of the approach, this could inadvertently introduce another layer of overfitting if the focus of the approach were on optical flow and self-motion estimation as it is here.
> We should also note that we use the term “navigation” in this paper to refer in a general sense to tasks involving attitude and position estimation, as commonly defined for typical Guidance-Navigation-Control (GNC) systems, which we have clarified by writing at lines 21-23:
> “Optical flow has also been used by drones for navigation tasks involving estimation of attitude and position, using inefficient and constrained algorithms whose application is typically limited to simple tasks such as holding position in hover.”
>
> End Response 2

---

> ### Author Response · Authors · 2022-08-22
> **Response 3**
>
> Beginning Response 3
>
> ## 5. [Why dynamically meaningful self-motion?]:
>
> We thank the reviewer for pointing out the need for clarification on the “dynamically meaningful motions” we defined. Self-motion produces coherent optical flow patterns specific to the motion that the agent experiences, and the structure of its surrounding environment. We introduced the term “dynamically meaningful motions” to distinguish FlyView from other optical flow datasets that only contain coherent motion of foreground objects (e.g. FlyingChairs), random motions not associated with any particular self-motion (e.g. MPI-Sintel), or mainly forwards motion typical of road vehicles (e.g. KITTI). Moreover, whereas rotational self-motions produce optical flow fields that depend only on the agent’s angular velocity, translational motions result in optical flow fields that blend information on the viewer’s velocity and position relative to the surrounding environment. As a result, global optical flow information can be helpful in two different ways. First, it gives cues essential for solving local motion ambiguities, such as those experienced with the aperture problem. Second, the global motion can help to remove or positively introduce bias learned during the training stage.
> When targeting optimal performance on a selected vehicle, it will clearly be necessary to match the network’s training to the specifications of the specific vehicle. In particular, it is preferable to train a network on the entire spectrum of dynamics that the vehicle will experience. For example, in the particular case of the fly, the visual system is responsive to specific optical flow patterns matching the fly’s dynamics (Taylor, G.K., & Krapp, H.G., 2007. Adv. Insect Physiol., 34, 231-316.  https://doi.org/10.1016/S0065-2806(07)34005-8.). On the other hand, it is reasonable to assume that a network targeting recognition of general self-motions will perform best if trained on a broad range of self-motions, as opposed to random optic flow fields. We therefore included the coherent optical flow fields resulting from random self-motions in the "Motion Primitives" part of the FlyView dataset. The pipeline that we present in our paper may of course be used to generate optical flow data in response to other dynamically meanginful motions if required.

---

> ### Author Response · Authors · 2022-08-22
> **Reponse 4**
>
> ## 6. [Applications]
>
> We thank the reviewer for their constructive comments on the applications of our research activities. We agree that this data will particularly benefit drone navigation when the dimensions of the drones are reduced significantly, down to the size of insects. Whilst this is not currently the case, micro-drones are being developed rapidly, and their dimensions continue to decrease. Our FlyView dataset may therefore prove useful to enabling better performance of camera-based systems and in reducing the number of sensors required to make an insect-scale drones feasible (see response to Additional feedback # 2 below). Not only can the FlyView dataset be used for work related to small drones, it also opens the door to a better biological understanding of the fly's visual system. In particular, it permits us to simulate the neuronal response to specific luminance signals captured by the photoreceptors, and to follow their propagation down the neural circuitry where optical flow is computed. We agree with the reviewer that the 4 mm stereo baseline is of limited interest in estimate the depth using disparity, and the fly itself calculates the depth in a structure-to-motion fashion, as discussed at lines 168-169 of the manuscript. However, the 4mm baseline is of interest in the biological analysis of the optical flow since the fly's visual system shares the optical flow signals between the two hemispheres.
>
> We decided to limit data collection to a 340˚ field of view for two main reasons, as explained at lines 171-173. First, the Blender-based simulator generates a singularity in the optical flow map at the discontinuity between 0˚ and 360˚ azimuth in our virtual camera system. Second, the horizontal field of view of the fly is slightly less than 340˚ (see Fig. 1A), and is therefore well modelled by our setup.
>
>
> ## Clarity:
>
> We thank the reviewer for suggesting ways in which the applications and importance of our FlyView dataset could be clarified, which we believe are now covered by our responses to their specific comments 2, 5, and 6 above.
>
>
> ## Relation to prior work :
>
> We thank the reviewer for suggesting improvements to our comparison with previous work. We have amended original lines 166-169 to read:
>
> “Typical stereo vision systems have a baseline from tens of centimeters (HD1K 0.3m, KITTI 0.54m) to more than one meter (FlyingThings3D, Monkaa, Driving), enabling accurate depth extraction over distances of tens of metres. In contrast, the 4 mm baseline between the compound eyes of the fly is too small for the binocular disparity to be used to extract useful depth information, except at  distances much less than 1 m. This baseline displacement may nevertheless be relevant when modelling downstream fusion of binocular information in flies (e.g. when analysing structure-from-motion), so is retained accordingly.”
>
> We thank the reviewer for their insightful comment on aerial/drone path planning in the computer graphics literature. We can indeed foresee scenarios in which it could be helpful to consider the physical limitations of the platform as suggested, and aim to collect such data in future works to test our deep learning models in real-world situations. Regarding the current dataset, drone trajectories were captured using MoCap because this was straightforward using techniques and equipment that were already available in our research lab, and allowed rapid iteration of the data collection whilst also supplying results of high known accuracy.

---

> ### Author Response · Authors · 2022-08-22
> **Response 5**
>
> ## Additional feedback:
>
> - As noted, the total duration of the blowfly flight trajectories is less than 1 s, but this is already sufficient to capture a broad range of flight behaviour. For instance, the blowfly Calliphora can reach turn rates of up to 1700˚ s-1 with accelerations of 3 g or higher, at a wingbeat frequency of approximately 150 Hz. The correspondingly high frame rate used in the renderings enables us to capture numerous manoeuvres that happen in a split second across a significant number of trajectories. New Section B3 in Supplementary Materials illustrates the high dynamics of flies in the small trajectories we used.
>
> - As the reviewer notes, current drones have a suite of inertial and other sensors contributing to their navigation. As an example, the DJI Tello drone that we used in our data collection includes a camera, an inertial measurement unit, and an ultrasound range detector for altitude estimation. Self-motion estimation relies heavily on sensor fusion to reduce the uncertainty of each individual sensor – particular in the case of accelerometry where shaking loads introduce significant vibrational noise, and where the high acclerations experienced during manoeuvres and magnetic field disturbances indoors make reliable attitude estimation a challenge. Performing the navigation task using a unique source of visual information, such as a camera, would therefore bring great benefits at small scales, provided that the algorithms are robust enough to reduce the uncertainty of the measurements. Indeed, optical flow has already been trialled as a replacement for inertial sensors [1, 2], thereby avoiding the adverse effects of platform vibration. Finally, because our dataset uses large field-of-view images, we increase the likelihood of having the direction of motion lying in the camera field of view, leading to a less noisy self-motion estimation compared to existing datasets.
>
> - We thank the reviewer for identifying this typo, which we have corrected.
>
> ## Additional references:
>
> [1] Expert F, Ruffier F. Flying over uneven moving terrain based on optic-flow cues without any need for reference frames or accelerometers. Bioinspir Biomim. 2015 Feb 26;10(2):026003. doi: 10.1088/1748-3182/10/2/026003. PMID: 25717052.
>
> [2] Fuller Sawyer B., Karpelson Michael, Censi Andrea, Ma Kevin Y. and Wood Robert J. 2014 Controlling free flight of a robotic fly using an onboard vision sensor inspired by insect ocelli. J. R. Soc. Interface.112014028120140281http://doi.org/10.1098/rsif.2014.0281.

---

> > ### Comment · Reviewer_6z2S · 2022-08-29
> > **Thanks for incorporating the comments to improve the paper**
> >
> > Thanks for your careful revision to incorporate the comments, especially by adding the new experiment for the optical flow estimation. I appreciate the authors' effort to improve their work. Therefore, I increase the score and support this paper acceptance.
> >
> > I really like the general direction of this research, and I believe using the wide field view cameras will help navigation in future. However, I still believe that this research can improve a lot, as described in below. These items can be discussed as future work directions:
> >
> > **1. [Limited virtual environments]** In general, I think using 9 virtual environments (only 2 outdoor scenes) is limited and might not be a good representative of the real-word applications. I think one potential reason for having few outdoor scenes is that authors used predefined paths and in an outdoor environment, there are more objects (e.g., trees) to interact with. This might make it harder to place the predefined paths in the virtual environment without having any collisions with the objects in the scene.
> >
> > **2. [Limited to static scenes]** The dataset is limited to static scenes and needs to add some dynamic scenes to be a better representative of the real-world applications.
> >
> > **3. [Limited pre-defined drone paths]** Few predefined drone paths are used, which is not representative of various possible drone motions. Various drone path shapes can be generated using the literature in drone path planning.
> >
> > **4. [Improving drone path data generation]**  I think only using dynamically meaningful motions might cause overfitting if the final goal is self-motion estimation for a similar reason to what authors mentioned in their reply: “ if the navigation dynamics were the focus of the approach, this could inadvertently introduce another layer of overfitting if the focus of the approach were on optical flow and self-motion estimation as it is here.” When the final goal is the self-motion estimation, having more variety in camera path shapes (e.g., splines with various shapes) provides a richer dataset because it covers a wider range of possible camera motions for the self-motion estimation. I think final data should have both dynamically meaningful motions as well as some randomness in the generated camera paths to avoid overfitting to the predefined paths used in this dataset.
> >
> > **5. [Training a self-motion estimator using this data and comparing it with onboard drone sensory data]** I would like to see an experiment comparing self-motion estimation using this method vs onboard drone sensors to see how much this data can improve self-motion estimation in real applications. For the real application, it is not easy to attach two cameras to drones, or there are not many drones with 360 cameras. Therefore, it will be super useful if authors show how their data can improve existing self-motion estimators for drones in the real-world application.
> >
> > **6. [Imitating insect fly and its interaction with its environment]:** What makes insects interesting is their interaction with complex environments and their ability to fly and navigate in complex scenes with respect to objects in their surroundings. However, the camera trajectories are designed without considering the interaction with the environment. Some predefined camera trajectories are rendered in different virtual environments. This dataset can be more useful, if one could use this data to imitate the insect path planning with respect to objects in a complex environment.
> >
> >
> > For drone path planning, there are many related studies that are missing in the paper (see below). Authors can use these papers to cover a wider range for drone camera motions for their dataset generation because one important application of this paper might be drone self-motion estimation in these studies:
> >
> > 1. “An interactive tool for designing quadrotor camera shots”, TOG 2015
> > 2. “Generating dynamically feasible trajectories for quadrotor cameras”, TOG 2016
> > 3. “Real-time planning for automated multi-view drone cinematography”, TOG 2017
> > 4. “Optimizing for aesthetically pleasing quadrotor camera motion.”, TOG 2018
> > 5. “Creating and chaining camera moves for quadrotor videography”, TOG 2018
> > 6. “Capturing subjective first-person view shots with drones for automated cinematography”, TOG 2020
> > 7. “Optimization-based user support for cinematographic quadrotor camera target framing”, CHI 2021
> >
> > I wish to see follow up papers using this dataset that improve self-motion estimation for drones.

---

### Meta-Review · Area_Chair_52pp · 2022-09-09

**Recommendation:** Accept
**Confidence:** 3

**Metareview:**

Accept(Poster), The paper received reviews leaning towards accept. The rebuttal by authors addressed a number of concerns, it would be useful to see improvements to the documentation of the dataset in the final version of the paper.

---

### Decision · Program_Chairs · 2022-09-16

Accept